# Comparison of landslide forecasting services in Piemonte (Italy) and in Norway, illustrated by events in late spring 2013

Graziella Devoli[1,2], Davide Tiranti[3], Roberto Cremonini[3], Monica Sund[1], Søren Boje[1]

1 Norwegian Water Resources and Energy Directorate (NVE), Section for forecast of flood and landslide hazards, Oslo, 0368, Norway

2 Department of Geosciences, University of Oslo, Oslo, 0316, Norway

Regional Agency for Environmental Protection of Piemonte (ARPA Piemonte), Department of Natural and Environmental Risks, Torino, 10135, Italy

*Correspondence to*: Graziella Devoli (gde@nve.no)

**Abstract**

A few countries in the world operate systematically national and regional forecasting services for rainfall-induced landslides (i.e. debris flows, debris avalanches and shallow slides), among them: Norway and Italy. In Norway, the Norwegian Water Resources and Energy Directorate (NVE) operates a landslide forecasting service at national level. In Italy the Regional Agency for Environmental Protection, ARPA Piemonte, is responsible for issuing landslide warnings for the Piemonte region, located in northwestern Italy. A daily hazard assessment is performed, describing both expected awareness level and type of landslide hazard for a selected warning region. Both services provide regular landslide hazard assessments founded on a combination of quantitative thresholds and daily rainfall forecasts together with qualitative expert analysis. Daily warning reports are published at http://www.arpa.piemonte.gov.it/rischinaturali and www.varsom.no.

On spring 2013, the ARPA Piemonte, and the NVE issued warnings for hydro-meteorological hazards due to the arrival of a deep and large low-pressure system, called herein "Vb cyclone". This kind of weather system is known to produce the largest floods in Europe. Less known is that this weather pattern can trigger landslides as well.

In this study, we present the experiences acquired in late spring 2013 by NVE and ARPA Piemonte. The Vb cyclone influenced the weather in Europe, and in both countries, for long time since the end of April until the beginnings of June 2013. However, mayor affectations were observed in the first half part of this period in Piemonte, while in Norway, major damages were reported across a period from 15th May to 2nd June 2013. Floods and landslides produced significant damages to roads and railways along with buildings and other infrastructure in both countries.

This case study shows that large synoptic pattern can produce different natural hazards across Europe, from sandstorm at low-latitudes, to flood and landslides when the system moves across the mountain regions. These secondary effects were effectively forecasted by the two landslides warning services, operating across Europe. The landslide risks were also properly

communicated to the society with some days in advance. This analysis has allowed to establish a fruitful international collaboration between Arpa Piemonte and NVE and the future exchange of experiences, procedures and methods under similar events.

# 1 Introduction

One of the targets proposed by the Sendai Framework (UN, 2015) is to substantially increase the availability of access to multi-hazard early warning systems and disaster risk information to people by 2030 (UN, 2015). It also emphasizes that there is need for enhancing preparedness, response, recovery, rehabilitation and reconstruction, in order to prevent natural disaster risk and that, response actions must be focused within and across sectors, by States, at local, national, regional and global level.

UNISDR (2009) defined an early warning system (EWS) as "a set of capacities needed to generate and disseminate timely and meaningful warning information to enable individual, communities and organization threatened by a hazard to act appropriately and in sufficient time to reduce the possibility of harm or loss". To be efficient, EWSs should include four elements: the knowledge of the physical mechanisms that cause the hazard and the exposed elements at risk; the technical capacity to continuously monitor the hazard and to develop changing scenarios to issue warning; the communication of the warning and the capacity to translate the prediction into warning and action plans (Cloutier et al 2015). Worldwide there are many EWSs currently operated for various types of natural hazards, including landslide hazards.

Landslides represent a serious hazard in many countries, causing yearly significant loss of lives (Petley 2012; Haque et al., 2016) and large damages to infrastructures. Landslides are defined as "the movements of a mass of rock, debris or earth down a slope" (Cruden and Varnes, 1996) and are classified based on the failure mechanisms and the type of material (Hungr et al., 2014). However, other parameters like rate of movement (e.g. velocity), size, depth of sliding surface, among others, can be used to classify them. Using the velocity as a criteria, the so-called "rapid landslides" are those with velocity >1.8 m/hr, and being "extremely rapid" the ones with velocity larger than 5m/sec (Cruden and Varnes, 1996). It is sometime useful to classify landslides based on type of triggering factors, thus expressions like "earthquake-induced landslides", "rainfall-induced landslides", "precipitation-induced landslides", "weather-induced landslides" and "snowmelt-induced landslides" are often used in literature (Baum and Godt 2001; Calvello, 2017; Katsura et al 2008; Rodriguez et al 1999; Havenith et al., 2016). The landslides triggered by intense rainfall, both short- or long-duration, are in general called "rainfall-induced landslides" or "precipitation-induced landslides, while if abundant snowmelt is causing them, "snowmelt-induced landslides" is often used. The expressions "weather-induced landslides" or "rainfall- snowmelt-induced landslides" has been used as general term to include both landslides triggered by rainfall and/or by snowmelt, especially in mountainous areas covered by snow, where they can occur simultaneously especially during spring.

The landslide types triggered by rainfall and snowmelt episodes are usually in the category of slide- and flow-type landslides (Hungr et al., 2014). The following types are commonly observed:

a) soil slides (e.g. clay/silt planar slides) and debris slides (e.g. gravel/sand/debris slides) are usually of small size (volume inferior to 5000 m$^3$ ), shallow slides with a sliding surface 1-2 m deep that occur within the soil material or at contact with the underlying less permeable bedrock (as also observed by Zêzere et al., 2015). They are planar slides; however, a few rotational ones may occur, especially in clay/silted soils.

b) debris avalanches occur often in open slopes, initiating as shallow planar soil- or debris slide.

c) debris flows and debris flood occur usually in steep channels, starting as stream bed erosion or by a soil slide, debris slide or debris avalanche from a steep bank, entraining material downslope.

See Hungr et al., (2014) for more details.

These  main types of soil slip develop in steep slopes and are characterized by high rate of movement, varying from rapid to

extremely rapid events. They occur in different types of soil, residual soils, colluvial, pyroclastic, fluvial and tills deposits or organic soils. Close to infrastructures and to buildings soil slides may occur in artificial loose fills. They are triggered by short duration rainfall events (minutes to hour), due to rapid infiltration and percolation of water in the thin soil material.

Debris flows are known to be the most destructive ones, because of their high velocity and long run-outs; debris avalanches are also quite destructive as they usually occur in clusters and due to their potential to spread out in the depositional area. Soil-

and debris slides, even if relatively shallow and of small size, occur in clusters, causing significant damages to infrastructures and even loss of lives if they occur close to inhabited areas. They occur over a large area all at the same time and often simultaneously with floods making damages much more extensive.

The occurrence of these types of landslides has become more frequent and, as population and infrastructures have increasingly expanded into landslide-prone areas, their impacts on society have become more dramatic. Recent studies shows that these

could be potentially enhanced under a changing climate (Stoffel et al., 2014; Gariano and Guzzetti, 2016 and reference therein). Effective landslide warnings have become essential elements of integral risk management worldwide, since they are a cost-effective risk mitigation measure and in some regions the only suitable option for a sustainable landslide risk management (Glade and Nadim, 2014). EWSs for landslides are designed to detect events that precede a landslide in time to issue an imminent hazard warning (Di Biagio and Kjekstad 2007) and initiate actions to mitigate and to reduce the potential damages

and allow people to get to safety.

The development of landslide EWSs has not been uniform worldwide, and a few public resources have been invested in the past for their establishment, probably because the landslides losses are perceived as private economical losses, like in USA (Baum and Godt, 2010). Apart from the case of Hong Kong, where the first landslide EWS was organized in 1977 and still operative since then (Chan et al., 2003), in other countries, is at the end of the 1990s that most of the EWS started to be

developed (D'Orsi et al., 1997).  In the 1980s, in USA, there were two attempts of EWSs. The first recorded debris flow early warning attempt was done in the spring of 1984 in the state of Utah, while in The San Francisco Bay area of California the first experimental operating landslide EWS started in 1985, but a decade later were closed, because of shortage of personnel and lack of adequate funding (Baum and Godt, 2010).

EWSs are technical feasible for some types of landslides. Overview and classification of existing landslides EWSs are presented in Thiebes et al, 2012, Bazin et al., 2012 and later Stähli et al. (2015). These last authors have proposed an overview and a classification of existing early warning systems for rapid mass movements (e.g. debris flows and snow avalanches) where three main categories are identified: (i) alarm, (ii) warning and (iii) forecasting. Another recent summary of existing weather-induced landslides EWSs is presented in Calvello (2017) in which systems are distinguished in *local* and *territorial*. The literature shows that many local early warning systems exist at specific sites, for rockslides, deep-seated complex landslides and debris flows where extensive monitoring instrumentation provides detailed information (i.e. Bardoux et al 2009; Blikra et al., 2013; Cardinaletti et al., 2011; Michoud et al., 2013 and references therein). The territorial EWSs have acquired importance in the last 15-20 years and especially after the Hyogo framework for Action (2005-2015), adopted by the World Conference on Disaster Reduction (UNISDR, 2005). They are mainly constrained to forecast the occurrence of rainfall-induced landslides (i.e. Osanai et al., 2010; Baum and Godt, 2010; Jakob et al., 2012; Liao et al., 2010; Pumo et al., 2015; Segoni et al., 2014; Lagormarsino et al 2013; Tiranti and Rabuffetti, 2010; Ponziani et al., 2013; Huang and Hong 2010).

Many countries have spent last decades working on preparing the technical basis for early warnings, by understanding landslide initiation, defining rainfall thresholds, installing real-time monitoring instruments and organize prototypes of landslide warning systems, but a few of them became operational. In agreement with other authors (Segoni et al., 2015) in general the potential of EWSs is not yet fully exploited by governments and decision-makers.

A broad range of literature exists on the definition of empirical rainfall thresholds for the possible landslide initiation or description of single components of the system (Segoni et al., 2018 and references therein). The thresholds may define the rainfall, soil moisture or hydrological conditions that when reached or exceeded are likely to trigger landslides. Therefore, successful prediction of landslide hazards in large regions greatly depends on ability to link meteorological conditions with various types and extents of slope failures. In the prevention of rainfall- and snowmelt induced landslides at territorial level, the recognition of the relationship between large-scale patterns and landslides occurrence is important to investigate. The synoptic weather and landslides occurrence has been demonstrated in a few works. Large-scale pattern phenomena such as the El Niño and the North Atlantic Oscillation (NAO) change slowly and impact in both precipitation regime and temporal occurrence of different landslide types in different areas of the world have been demonstrated. In 2005, Trigo et al. found connections between the North Atlantic Oscillation (NAO) index and the occurrence of landslides in Portugal. Wood et al. 2016 investigated the synoptic weather trends and landslide in the European Alps. The meteorological control on the distribution of debris flows in Iceland was also investigated by Decaulne and Sæmundsson (2007). Large synoptic weathers were investigated in southern Norway (Devoli et al., 2007). To understand the relationship between weather, climate, precipitation and landslide occurrence will allowed to predict in advance the possible development of certain type of weather and be better prepared.

Literature shows that few examples describe how warning systems work, their performance or how warnings are received by population or stakeholders. The comparison of different EWSs and specifically their operation is even rarer and a quite unexplored topic. To our knowledge a few have been made (e.g. Baum and Godt, 2010; Lagomarsino et al., 2015; Zêzere et

al., 2015) but they mainly compare some technical components, like the application and performance evaluation of different thresholds in the same area. The various EWSs consist of different components, and therefore it is important to analyse not only the performance of the single components, but the performance of the entire system. However, components like expert knowledge decision are difficult to judge in objective way and there are not methods in place. The lack of comparison among existing EWSs is also due to the fact that many EWSs are still not fully operative or they have recently started operation, and therefore lacking data for analysis.

Norway and Italy have a long tradition of flood forecasting, but only in (relatively) recent years, efforts are made to design, develop and operate landslide forecasting services, often in synergy with flood and/or snow avalanche forecasting. The Norwegian Water Resources and Energy Directorate (NVE) operates a landslide forecasting and warning service at national level (in Norwegian "Jordskredvarsling") since 2013. The service is relatively new. Since its beginnings, the attention has been on the design and the implementation of the service at national level, instead of describing its function to an international audience. The emphasis has been on the establishment and training of forecasters, on the development of existing web tools used for flood and snow avalanche forecasting to include landslide related parameters and thresholds, on the establishment of routines, implementation and updating landslide thresholds and definition of warning and performance evaluation criteria (Colleuille et al., 2017; Devoli et al., 2014; Boje et al., 2014). The performance of the service was recently tested using the Event, Duration, Matrix Performance (EduMaP) in Piciullo et al. (2017) and the description of the entire service is proposed in (Krøgli et al., in review 2017).

In Italy the landslide hazard assessment is not national but regional. In Piemonte the regional environmental agency (Regional Agency for Environmental Protection, ARPA) is responsible of the daily landslide hazard assessments and emission of landslide warnings and operated the local EWS since 2008 (Tiranti and Rabuffetti, 2010).

In this study, we compare two examples of forecast and warning services for rainfall and snowmelt-induced landslides successfully operating in Piemonte, north-western Italy, and in Norway. The main objective of this study is to shows how the two services are organized, but also how they operated and performed under the same large synoptic pattern that hit Europe on 2013.

Another important objective of this work is to demonstrate that the weather synoptic systems, known as Vb cyclones, are often responsible of intense rainfall events and the associated to high temperatures producing intense snowmelt in many European countries and triggering not only large floods but also many landslides. Quite often forecasting services focus on the analysis of the climatic and meteorological conditions in their own region, forgetting that the rainfall can be part of larger processes and landslide can occur across municipalities, regions and even countries at the same time. An example is the landslides triggered by Hurricane Mitch in 1998 across the Central America countries (i.e. Bucknam et al., 2001; Cannon et al., 2002) or the landslides triggered by the storm Desmond the 4[th] and 5[th] December 2015 in UK and in Norway. The study wants to demonstrate that cooperation between countries and between operational landslide warnings is necessary and the increased knowledge of weather system patterns and share experiences can enhance forecast leading time and improve daily landslide hazard assessement.

# 2 Study areas

Although the two areas are located at different latitude, both ones are characterized by complex orography and similar geologic surficial processes. Moreover, according to Peel et al. (2007), both Norwegian and Italian Alps belong to the same Köppen–Geiger climate class.

## 2.1 Piemonte region, Italy

The Piemonte region is complex from a geomorphological and geological point of view. Its territory is shaped by mountain environments (Western Alps and subordinate Apennine with peaks ranging from 1000 to 4800 m asl), hills (Torino Hill, Monferrato hills and Langhe with an elevation range of 400-700 m asl), and alluvial plains (200-300 m asl), surrounded by the

Alps and Apennines on three sides (**Fig. 1a**). The Western Alps are characterized by a complex double-verging structure with asymmetrical transversal cross-section (Roure et al., 1990, 1996; Pfiffner et al., 1997), subdivided into three main structural sectors. (1) Southalpine domain characterizes the Internal sector (the collisional system upper plate consists in Hercynian and pre-Hercynian bedrock formed by lower continental and upper mantle rocks). (2) Helvetic–Dauphinois domains constitute the External sector, representing the European foreland zone, formed by Hercinian intrusive massifs system and Mesozoic flysch

cover. (3) frontal thrust of Pennidic domain and the Insubric front (Malusà and Vezzoli, 2006) bound the Axial sector formed by Hercynian and pre-Hercynian continental rocks and Hercynian metasedimentary formations, oceanic lithosphere rocks and ocean fronting continental boundaries and orogenic flysch units. During the Quaternary, wide glaciers occupied alpine valleys and modeled by glacial pulsations. Locally, glacial landforms and deposits were modified by Holocenic fluvial/torrential processes, associated with widespread landslides (Soldati et al., 2006).

The geology of hilly environment is mainly formed by Oligocenic-Miocenic sedimentary strata, originated during the Tertiary Piemonte Basin where the lowest term of sedimentary sequence is formed by shallow-sea deposits, while deep marine environment (turbidite deposits up to 4 km thickness) represents the upper part of the sedimentary sequence. The stratigraphic succession is due to Oligocene marine transgression made by the alternation of marls, sandstones and shales (about 5-50 cm thickness); strata dipping NW with 8°-15° inclination. Sin-sedimentary tectonics controlled the thickness and lateral

interdigitations of the stratigraphic successions. The northward movement of the Padan thrust belt (Falletti et al. 1995) caused the progressive uplifting of the basin followed since Langhian. The sedimentary sequence lies on alpine metamorphic units by unconformity (Biella et al. 1987, 1992; Gelati and Gnaccolini 1988). Appennine units are poorly represented within the borders of Piemonte, mainly represented by Ligurian, Subligurian and Epiligurian units (**Fig. 1b**).

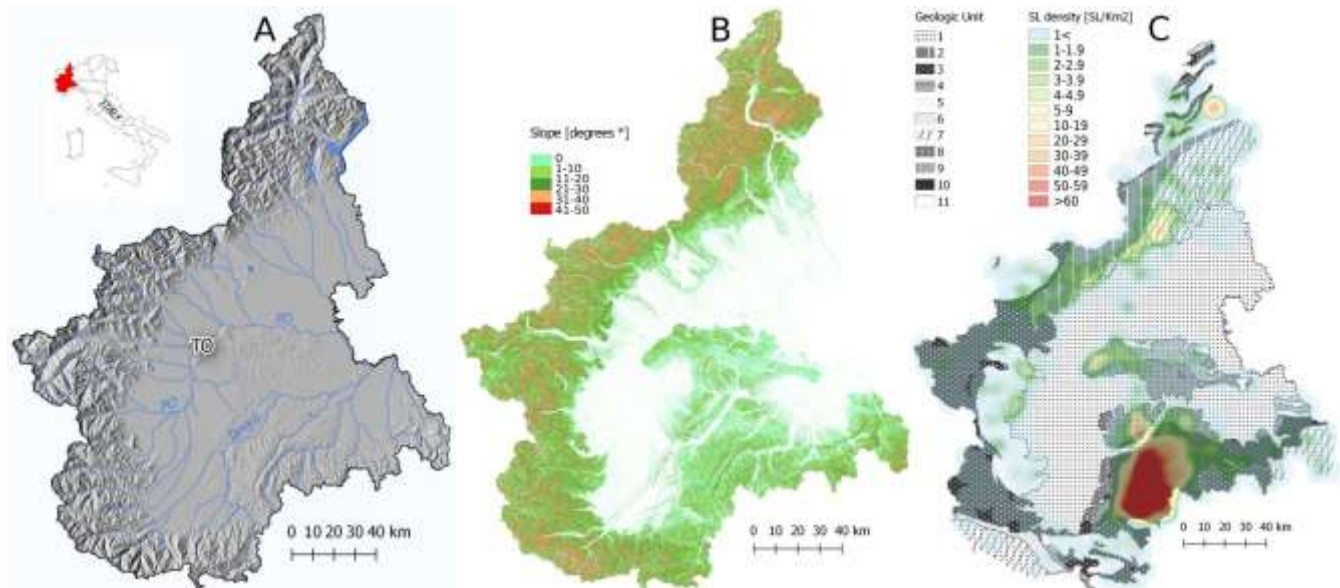

**Fig. 1:** A) Physiography of Piemonte, "TO" stands for Torino; B) Slope distribution of Piemonte; C)Density distribution of shallow landslides (from 1962 to 2016) compared with the geological/structural sketch map of Piemonte: 1. Quaternay; *ALPS*: 2. Austroalpine domain (pre-Alpine crystalline basement and Palaeozoic cover); 3. Pennidic domain (Permian–Mesozoic–Tertiary metamorphic cover; 4. Pennidic domain (Helminthoid Flysch Units); 5. Penninic domain (pre-Triassic crystalline basement); 6. Helvetic domain (Permian–Mesozoic cover); 7. Helvetic domain (pre-Alpine crystalline basement and Carboniferous cover); *APENNINE and HILLS:* 8. Internal margin foredeep deposits; 9. Epiligurian Sequences (episutural basins deposits unconformably covering the Ligurian units); 10. Epiligurian Sequences ("Oligo-Miocene" of Langhe); 11. Ligurian and Subligurian Units (nappes, locally ophiolitic-bearing); a. Front of tectonic units (limits of different paleogeographic domains); b. Neotectonic deformation zones.

As shown in **Fig. 1c**, the main occurrence of shallow landslides (density interpolation map of 33000 shallow landslides occurred from 1962 to 2016) is in correspondence of the Epiligurian Sequences ("Oligo-Miocene" of Langhe - hilly environment), the Helvetic domain (pre-Alpine crystalline basement and Carboniferous cover - Northwestern Alps), the Penninic domain (pre-Triassic crystalline basement - Northern Alps), the Internal margin foredeep deposits (hilly environment) and the Epiligurian Sequences (episutural basins deposits unconformably covering the Ligurian units - Torino Hill).

The spatial distribution of annual rainfall shows high precipitation in the northern areas with more than 2,100 mm per year, and low in the eastern part of the plains with less than 700 mm per year (**Fig. 2**). The monthly distribution of precipitations in Piemonte shows a bimodal distribution, with two peaks during spring and fall, and two minimums, during winter and summer. Four rainfall regimes (three continental ones and one Mediterranean) can be distinguished;

- Prealpine: dry season during winter, main maximum during spring and secondary maximum during fall;
- Subcoastal: dry season during summer, main maximum during fall and a secondary maximum during spring;
- Subalpine: dry season during winter, main maximum during fall and secondary maximum during spring;
- Subcontinental: dry season during winter, main maximum during fall and a secondary maximum during summer.

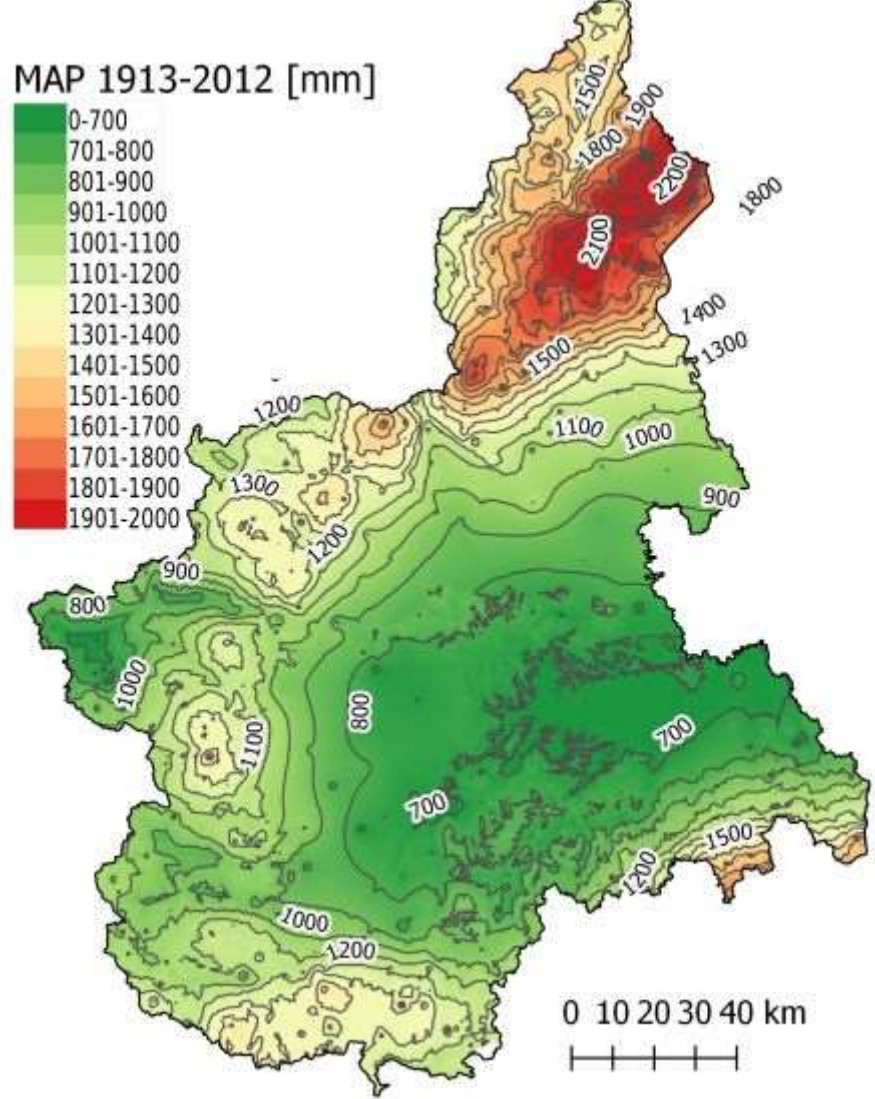

**Fig. 2:** Mean Annual Precipitation (MAP) in Piemonte from 1913 to 2012 (source: Arpa Piemonte).

## 2.2 Østlandet region, Norway

5    The region Østlandet in southeastern Norway also has a complex geomorphological and geological setting. The region includes eight administrative counties. The highest mountains are in the northern and western part of the area with maximum elevations up to 2469 m. asl, observed in the Jotunheimen area. The region is mostly hilly, with dominant landforms represented by glacially scoured valleys directed N-S in the eastern sector, while NW-SE in the western sector that congregate on to the Oslofjord (**Fig. 3a**). The valleys Østerdalen and Gudbrandsdalen are the longest in the country. The region contains also some

very large areas of lowland surrounding the Oslofjord. The longest river and watercourses, Glomma, and the biggest lake, Mjøsa, are also located in this region. Southeastern Norway contains extensive areas with forest and rich arable land. From a geological point of view, this region is dominated by bedrock of the Baltic shield, characterized by Precambrian basement rocks (e.g. granites, granodiorite, gneisses, amphibolites, rhyolite, gabbro, diorite and meta-sediments). In the northern sector rocks within the Caledonian orogen (e.g. sandstone, schist, amphibolite, micaschist, phyllite conglomerate) prevail. Cambro-Silurian sedimentary rocks (e.g. shale, limestone, phyllite) and Permian volcanic rocks (syenite, granite monzonite, porphyritic rocks and basalt) occur within the Oslo Graben (Solli and Nordgulen, 2006). Quaternary deposits that cover the bedrock are mainly left by glacial processes. Continuous till deposits cover large areas of the hilly mountains and valley sides and floors, with a variable thickness from 0.5 to a couple of meters. The bottom of the valleys is mainly covered by thick fluvial and glaciofluvial deposits. Till deposits have a large heterogeneity in terms of granulometry and composition. The amount and the composition varies as function of the bedrock, in some places the till deposits are covered by landslide deposits occurred after glaciation. Marine clay deposits are observed in the southern sector of the region.

Most of the rainfall-induced landslides in the region occur in proximity of steep slopes, (**Fig. 3b**) covered by till deposits, especially where there is a large clay mineral that reduce the water infiltration and provide more surficial runoff. The red areas in **Fig. 3c**, show were landslides susceptibility is high in the region. Two sectors are most prone to landslides: the steep slopes of N-S and NW-SE oriented glacially scoured valleys, covered by till deposits, where mainly debris flows and debris avalanches are observed, and the southern sector, where marine clay deposits are prevalent. Here clay slides may form and also quick clays slides, these triggered mainly by human activities.

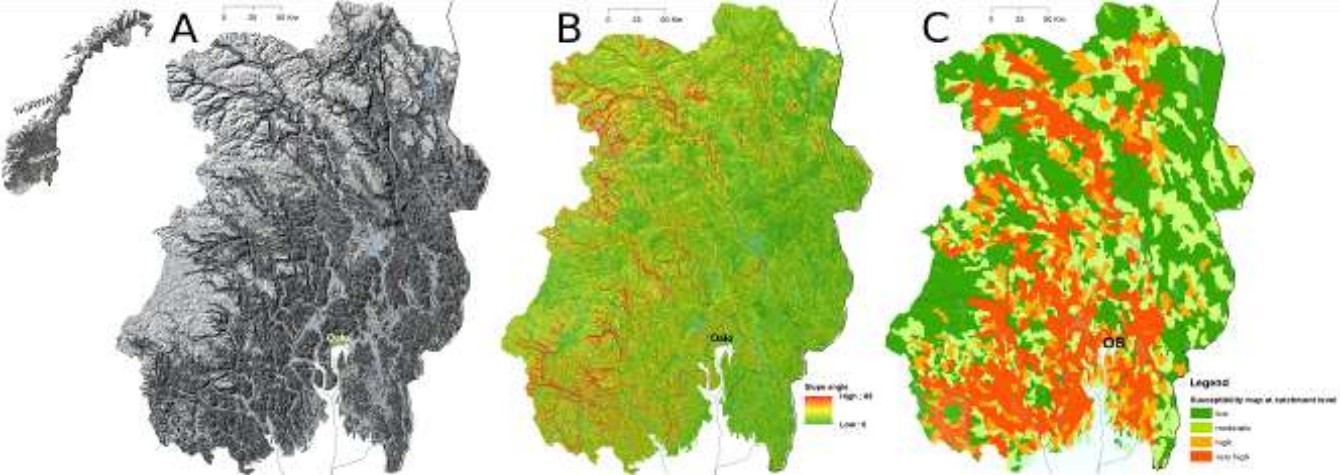

**Fig. 3:** Østlandet region: A) Physiography; B) Slope distribution; C) Landslide susceptibility map (unit: 1st order catchment).

Based on Köppen classification, the climate of the region is varying from a Tundra type (ET) in the Northwestern part, to Subartic (Dfc) in the central part. The Warm-summer humid continental type (Dfb) and Oceanic type (Cfb) are mainly observed in the southern sector and along the southern coastline. In this region, the climate is mainly characterized by cold winter and warm summer. The amount of precipitation in form of rainfall and snow varies depending on the area (i.e. valley floor and

mountain), but in general this area is the driest of Norway with low precipitation and mostly during summer. Deficit of precipitation are observed at Skjåk where the annual precipitation is of about 317 mm (water equivalent) or Biri with 754 mm annual precipitation. The area has a normal stable snow cover during winter, with normal annual maximum (1971-2000) around 1000 mm in the mountain (**Fig. 4**). The annual medium temperature ranges from -7°C in the mountain area to 7°C in the coastal area.

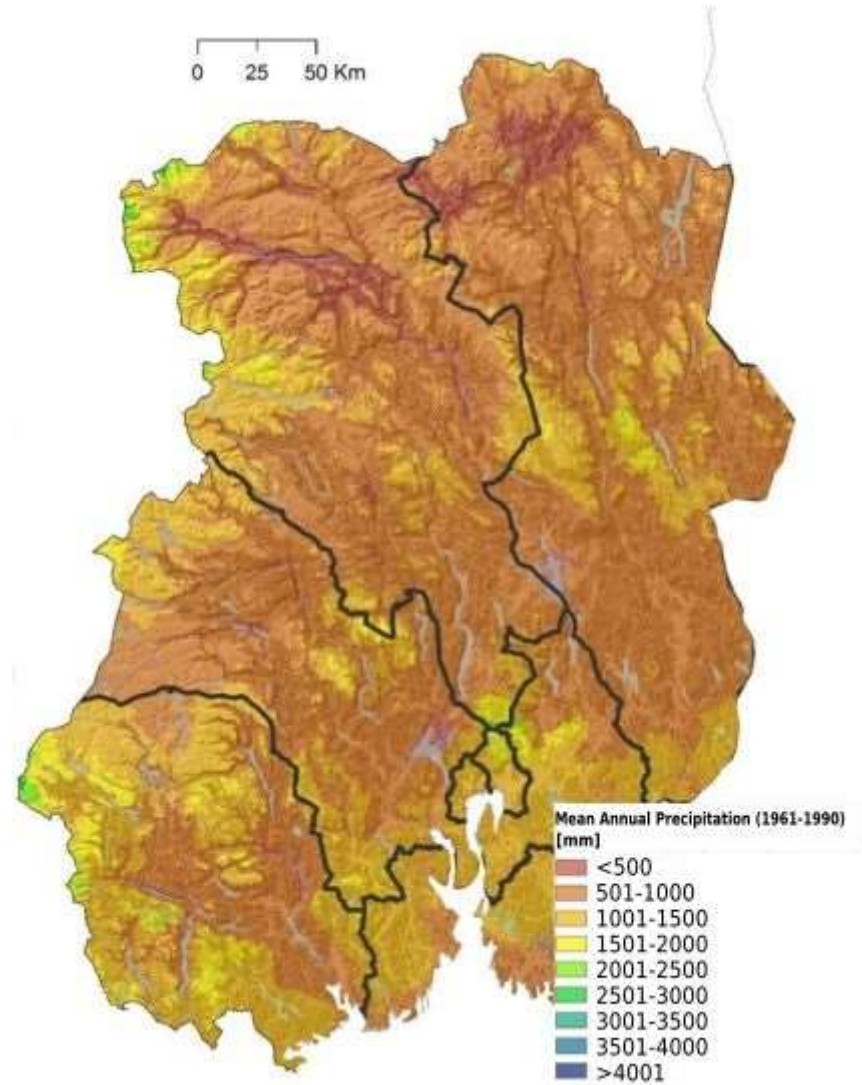

**Fig. 4: Mean** Annual Precipitation (MAP) of Østlandet region.

# 3 The landslide forecasting services in Piemonte region and in Norway

The warning services herein presented are classified as "territorial" (Calvello, 2017) and they can be classified as both "forecasting-type" and "warning-type" services (Stähli et al., 2015), because they predict the level of danger, and the occurrence of multiple landslides over a warning area and at regular intervals (e.g. daily), but also because data interpretation and the initial alert are based on predefined thresholds. Experts consult prediction models, thresholds and they analyze observations to forecast the regional danger levels, which are communicated widely in a bulletin. The main goal of services is to save lives, to reduce landslide risk for roads, railways and settlements, also increasing safety and predictability. In addition, to contribute to a better foundation for emergency preparedness at local level, services provide continuous information on conditions and expected development to national and regional authorities and the public.

As stated in section 1, both the services are designed to forecast the occurrence of rainfall- and snowmelt induced landslides, i.e. shallow landslides and debris slides, debris avalanches and debris flows (Hungr et al., 2014); the Norwegian service is alse responsible for slushflows warning. **Table 1** summarizes the main characteristics of the services herein described, showing similarities and differences.

**Table 1** – Characteristics of the EWS from Arpa Piemonte and Norway.

| EWS | Piemonte, Italy | Norway |
|---|---|---|
| Operators | Arpa Piemonte (Regional government) | NVE (National government) |
| Activated | 2008 | 2013 |
| Status | Operative (daily) | Operative (daily) |
| Landslide types | Shallow translational slide; channelized debris flow; rotational slide | Shallow translational slide; channelized debris flow; debris avalanches; slushflows |
| Type of triggering | Rainfall and snowmelt | Rainfall and snowmelt |
| Thresholds for different type of landslides | Yes | No |
| Rainfall/water input | Rainfall from rain gauge and weather radar | Water supply (rain + snowmelting) and soil moisture, interpolated from HBV model |
| Type of thresholds | ID thresholds for Alpine and Hilly environments for shallow slides; Radar hourly intensity rainfall thresholds for debris flows in alpine catchments; Antecedent precipitation thresholds for translational/rotational slides in hilly environment | Water supply vs degree of soil saturation Water supply vs degree of soil saturation + landslide susceptibility Water supply vs degree of soil mosture + soil frost |
| Methods for threshold definition | Statistical approach (Tiranti and Rabuffetti, 2010; Tiranti et al., 2013; | Statistical approach (Boje et al., 2014) |

|  | Tiranti et al., 2014) |  |
|---|---|---|
| Weather forecast | COSMO I7 NWP model: for the first three day with 6h resolution; Weather Radar QPE and Storm Tracking nowcasting | AROME MetCoOp model for the first 3 days as 1-$km^2$ raster maps, with a 24h and 3h resolution; EC model for the remaining 3 days as raster maps 1-$km^2$ resolution + daily briefing with meteorologist on duty +Visual inspection of radar during summer |
| Monitoring instruments | Multisensor weather gauges (<400); two weather radars | Multisensor weather gauges (~400); Groundwater level (80); water discharge (~ 350); other instruments (snow water equivalent; soil water content and soil temperature) |
| Released warning | Every day, before 1:00PM | Every day, before 11:00AM and updated before 3:00PM |
| Warning valid | From 1:00PM to 12:00AM | from 7:00AM the day of publication to 7:00AM the following day (8AM to 8AM Daylight Saving Time) |
| Warning days | 36 h (D0 and D1) | 3 first days (D0, D1 and D2) |
| Warning zone | Fixed (catchment) | Variable (county/group of municipalities) |
| Number of warning levels | 4 | 4 |
| Warning web page | http://www.arpa.piemonte.gov.it/rischinaturali | www.varsom.no |
| Broadcast media | Internet | Internet, CIM (Crisis Information Management) |
| Susceptibility map | 1:100000 scale (Tiranti and Rabuffetti, 2010); Alpine catchments (Tiranti et al., 2014) | Catchment level (Bell et al., 2014) 1:50.000 scale (Fischer et al., 2012) |
| Landslide database | >35000 landslides and debris flows https://webgis.arpa.piemonte.it/ | >57000 mass movements[1] www.skredregistrering.no |
| Landslide Verification after events | Web, social, newspapers and field observations | Web, regobs.no, newspapers and field observation |
| Primary references | Tiranti and Rabuffetti, 2010; Tiranti et al., 2013; Stoffel et al., 2014; Tiranti et al., 2014; Tiranti et al., 2016 | Colleuille et al., 2017; Piciullo et al., 2017; Krøgli et al., in review 2017 |

## 3.1 Piemonte's landslide forecasting service

The Regional Warning System for Geo-Hydrological hazards of ARPA Piemonte includes three independent early warning systems (EWSs), based on empirical rainfall thresholds and designed *ad hoc* for different typology slope processes, whose
5  triggering is generally determined by precipitations with different intensities over different durations:

---

[1] Landslides+ snow avalanches+ submarine landslides

- DEFENSE (*DEbris Flows triggEred by storms – Nowcasting SystEm*) is operated to forecast channelized debris flows occurrence in small alpine catchments. DEFENSE works by the intersection in GIS environment between alpine catchments, classified by the Clay Weathering Index, and instantaneous rainfall intensity (mm/h) provided by the weather radar using a storm-tracking algorithm (Tiranti et al., 2014). The use of quantitative precipitation estimations (QPEs) by the C-band weather radar is limited to alpine areas where the radar visibility is good and uncertainties limited. More details on the operational weather radar operated by ARPA Piemonte can be found in (Davini et al 2011; Cremonini and Bechini 2010).

- SMART (*Shallow landslides Movements Announced through Rainfall Thresholds*) is operated to forecast shallow landslides in mountain area (zone 1) and hilly environments (zone 2) Thresholds equations in the two zones are:

$$I = 25 \cdot D^{-0.45} \qquad \text{Zone 1} \tag{1}$$

$$I = 40 \cdot D^{-0.65} \qquad \text{Zone 2} \tag{2}$$

Where *I* is the rainfall mean intensity (mm/h) and *D* is the rainfall duration (h). More details are reported in Tiranti and Rabuffetti (2010).

- TRAPS (*Translational/Rotational slides Activation Prediction System*) is operated to forecast deep-seated translational and rotational slides in the hilly environment. TRAPS analyses 60-days antecedent precipitations, including water from snow melting (Tiranti et al., 2013).

ARPA Piemonte daily evaluates EWSs response to issue a regional warning to Civil Protection municipalities and citizens on slope processes occurrence. All the EWSs responses are displayed and managed through a WebGIS interface (**Fig. 5**) that allows a real-time estimation of hazard scenarios induced by observed and/or forecasted weather conditions. All the Piemonte EWSs are operative 24h/7d and automatic warnings are issued by e-mail and SMS to experts.

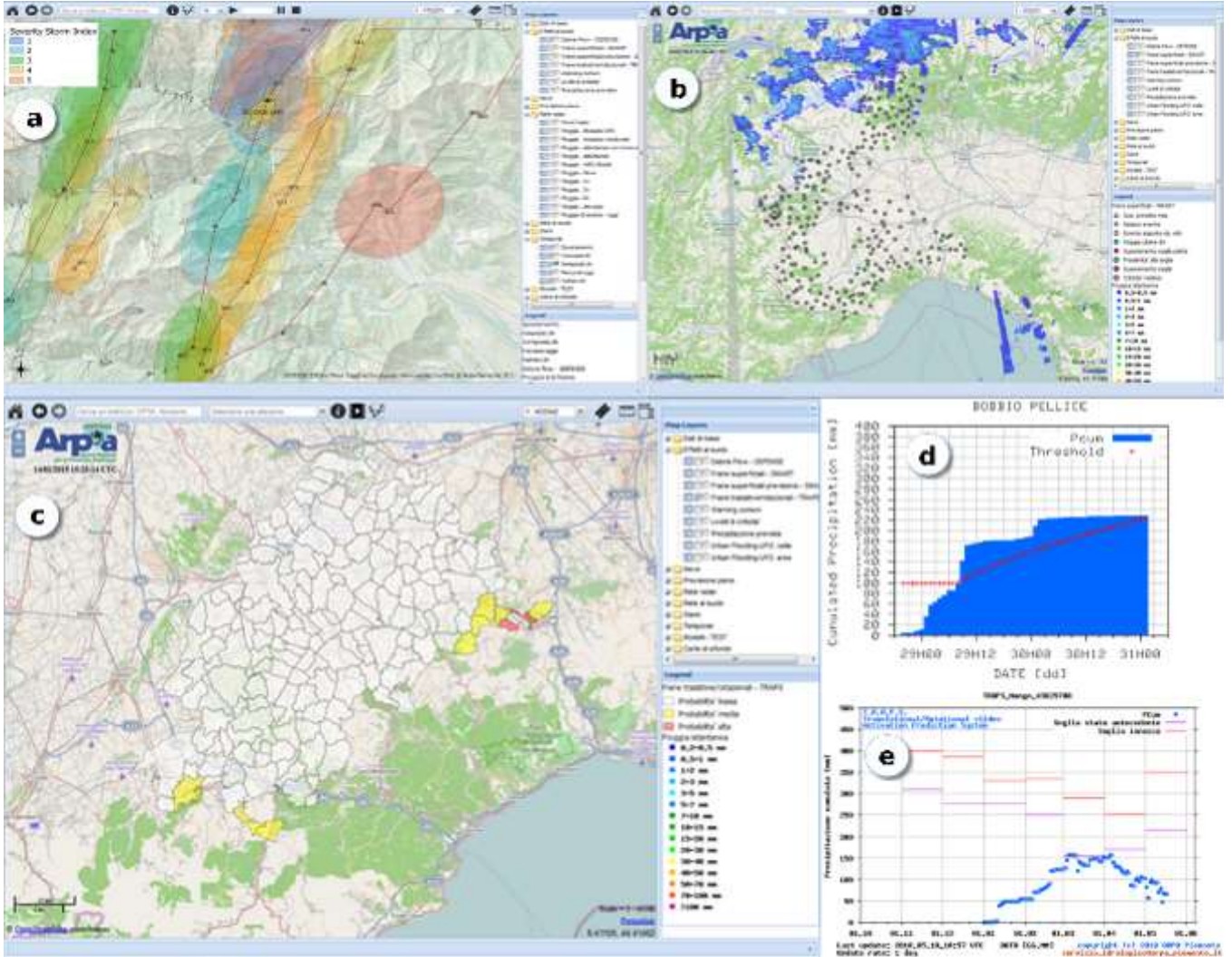

**Fig. 5:** Examples of EWSs WebGIS interface. a) DEFENCE: storm's cells are ellipses, lines are storms' path, yellow polygon is the catchment affected by debris flow triggering rainfall intensity; b) SMART: dots represent the rain gauges linked to shallow landslides triggering thresholds; c) TRAPS: polygons represent the areas characterized by different probability for translational landslides activation (white = low/null probability; yellow = medium probability; red = high probability); d) an example of SMART thresholds (red dashed line) representation related to accumulated rainfall (blue area) recorded by rain gauges; e) an example of TRAPS diagram: blue dots are the antecedent precipitation values accumulated over previous 60 days, red lines are the month triggering threshold value and purple lines are the monthly predisposing thresholds (thresholds indicating the high probability of early instability) (source: Arpa Piemonte).

## 3.2 The Norwegian landslide forecasting and warning service

The landslide assessment is done by a forecaster who daily consults the weather forecasts, the landslides thresholds and others relevant hydro-meteorological parameters. This information is available as raster data with 1-km$^2$ resolution and presented as

thematic maps at link xgeo.no, a web portal that assist experts in the daily forecast of floods, snow avalanches and landslides (Krøgli et al., in review 2017). Here, weather prognosis, forecasted thresholds and hydro-meteorological parameters, are published daily for the next six days. The portal also visualized past interpolated weather observations, thresholds and others parameters. Besides the thematic maps, the forecaster on duty may need to consults real-time hydro-meteorological

observations, in particular groundwater level or water discharge values at specific stations within the possible warning area. These data are also available in the same web portal xgeo.no.

The weather prognosis, temperature and precipitation, are obtained from AROME MetCoOp model (used in the Scandinavian regions as short-term forecast model for the first 3 days) and EC model (a global long-term European model, for the remaining 3 days). They are provided with a 24h and 3h resolution. Temperature and precipitation are also used as input variables in two

hydrological models (Krøgli et al., in review 2017). The main model is a distributed version of the conceptual HBV-model (Beldring et al., 2003) that divide the country in grid-cells, each one modeled as a separate basin with a corresponding water balance simulation. From temperature and precipitation as input variables, the model simulates forecasted hydro-meteorological variables like, rainfall and snow melting, water supply, degree of soil saturation, ground water level compare to normal, soil frost depth, water feed capacity, etc. In addition to HBV, a second tool is a physically based model, S-Flow

developed by NVE, that simulates water and heat dynamics in a column of layered soil covered by vegetation. The model uses also temperature and precipitation, but it requires also wind speed, relative air humidity and solar radiation as input data.

Unlike other countries, Norway does not use classical Intensity-Duration thresholds. Based on Guzzetti et al. (2008), the threshold used in Norway can be classified as "other thresholds", because they are based on analysis of historical landslides and water supply (e.g. rain and snowmelt) and the degree of soil water saturation (**Fig. 6a**). Both parameters are simulated

from HBV model (**Fig. 6b and 6c**). The first parameter is the water supplied to the soil from rain and snowmelt and is expressed as percent of yearly normal water supply in the reference period 1981-2010, and is the product of simulated snow melt and interpolated precipitation. The second parameter is the degree of soil saturation described as percent between the present soil water content compared to the maximum soil water content in the same reference period. The thresholds were derived using tree classification system. Generally, the thresholds indicate increased landslide hazard when values of water supply are higher

than 6-8% of the mean annual precipitation combined with a simulated soil water saturation degree higher than 60%. Three thresholds are used to separate conditions similar to warning levels of green, yellow, amber and red level (**Fig. 6a**). Since the thresholds were derived from tree-classification technique, the threshold consists of several linear equations generating thresholds with a latter shape. Thresholds are unique for all landslide processes. They were derived initially for the entire country, from few storms events in south Norway, but, recently, thresholds were defined including landslides events from

other regions (Boje, 2017). The spatial distribution of the thresholds is visualized as raster data (with 1-km$^2$ resolution) at xgeo.no (**Fig. 6d**). In order to consider issuing a warning the threshold map should displays a regional impact over a county or over a group of municipalities. Recently the thresholds has been combined with landslide susceptibility maps at catchment levels and a better thresholds is available that help to reduce the area of warning. Expert knowledge is fundamental in the daily

landslide hazard assessment, and, to decide the final assessment and extension of warning levels. An organization flow chart is presented in Piciullo et al., 2017.

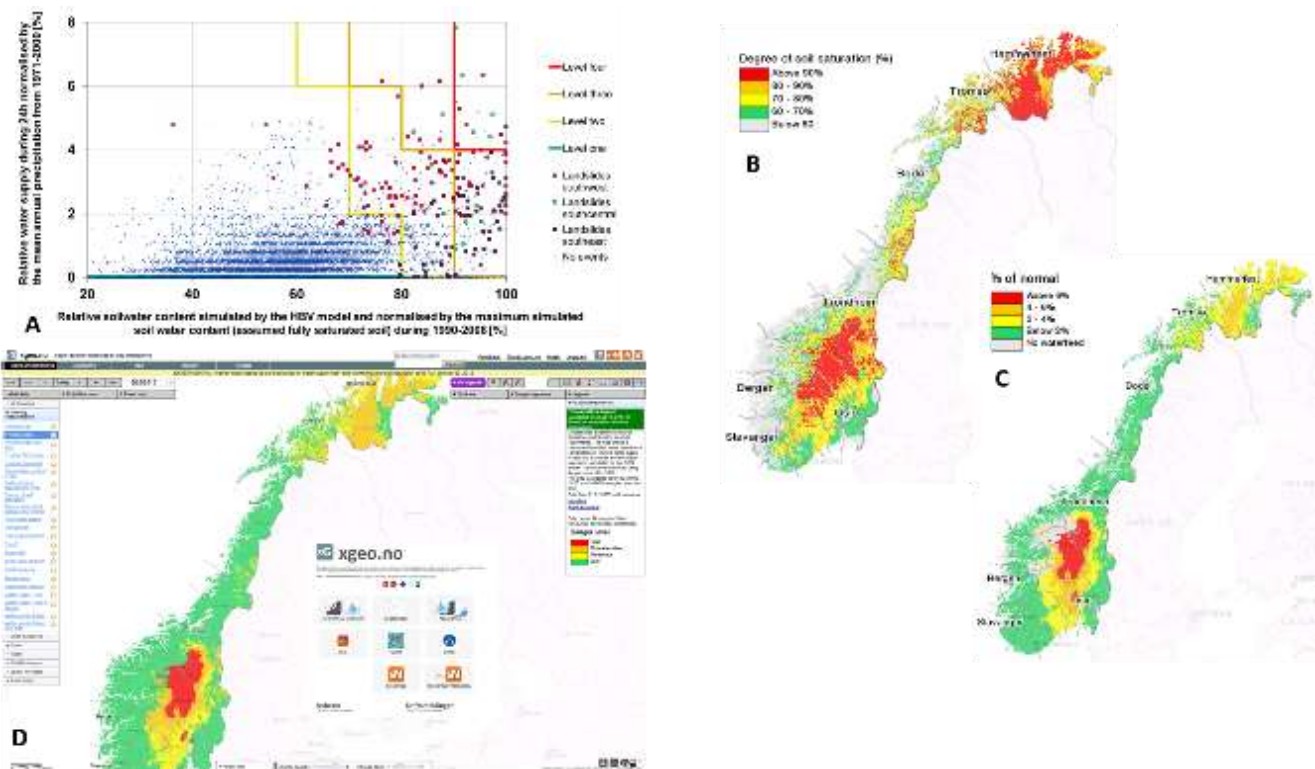

Fig. 6: Landslide thresholds and WebGIS interface xgeo.no. a) national landslide thresholds based on simulated degree of soil saturation and water supply obtained from HBV model; b) Map of simulated degree of soil saturation. The percent describes the relationship between todays soil water storage compared to the maximum soil water storage simulated with the HBV-model in the reference period 1981-2010; c) Map of simulated water supply (rain and snowmelt) the last 24 hours as percent of yearly normal water supply in the period 1981-2010, and is the product of simulated snow melt and interpolated precipitation; d) The web interface xgeo.no with the Hydmet landslide thresholds map in the background. The map represents of the national landslide thresholds presented in a) and, obtained combining the maps in b) and c). The maps in b), c) and d) are examples from the 22nd May 2013 and extracted from xgeo.no.

# 4 The Vb cyclones

Floods and landslides are important secondary effects of high-impact weather events, like tropical and extra-tropical cyclones, as they are accompanied by extremely strong winds and heavy precipitations. Central Europe and the northern Alpine region are exposed to of high-impact events associated to the Vb cyclones (Messmer et al., 2015). This type of cyclone was mentioned by Köppen (1881) and later, defined by Van Bebber (1882 and 1891) who proposed a cyclone classification based on the main

storm circulation trajectories in Europe (Messmer et al., 2015; Roald, 2008), describing one of them as Vb. In later classifications like the GWL/SVG classification proposed by James (2007) this synoptic weather regime is known as 11 TM «Tief Mitteleuropa=Low (Cut-Off) over Central Europe», while in the GWT classification is known as "TME Central European low".

The origin of Vb cyclones is either the Bay of Biscay, the Balearic or the Ligurian Sea, where moisture uptake occurs. The cyclone moves eastward over the southwestern part of France and over the Mediterranean Sea, where it refills with moisture and energy. Then, Vb cyclones move across Northern Italy and the Adriatic Sea before they turn northward to the Black Sea or Saint Petersburg, and finally to North-West towards Scandinavia (**Fig. 7**).

The Vb cyclones are characterized by very warm and humid air masses from the central Atlantic and Mediterranean, with cold
air masses linked to depressions in the northern part of the Atlantic forming a quasi-stationary front with extremely heavy rainfall (**Fig. 7**). The synoptic configuration is linked to blocking anticyclones in the North Atlantic and over Finland or the Kola Peninsula. This weather circulation occurs typical on July or August, but in recent years it has been observed in late spring (April, May and June).

Most of studies on Vb cyclones presented case studies of floods induced by Vb cyclones, focusing on source of moisture,
while few studies focused on analyzing the decrease or increase of number of cyclones. A description of the basic climatology of this weather type is provided in Messmer et al. (2015), given insight into the Vb cyclones variability and investigating their physical mechanisms.

These cyclones transport large amounts of atmospheric moisture to the central Europe and northerly side of the Alps, thus triggering extreme precipitation events (Messmer et al., 2015). The potential of transporting extreme precipitations to central
Europe is especially high if these cut-off low systems are positioned in the northern or eastern parts of the Alps (Awan and Formayer, 2016). There is agreement among authors on the large-scale dynamics of Vb events, which indeed seem to determine whether a Vb cyclone delivers high precipitation or not (Messmer et al., 2015). Even if they are rare events, 2.3 per year (Messmer et a., 2015), the Vb cyclones are highly relevant for Europe because of their potential to produce extensive precipitation and subsequent floods, particularly, during the warm season, and often in Austria, Switzerland, Germany, Poland
and the Czech Republic. The Vb cyclones are well known, among hydrologists and meteorologists, to have caused most of the largest floods in Central Europe, in the Elbe, Danube, or the Rhine catchments and in the Alpine area, like the: "1000-year flood" in 1342 in rivers Elbe, Danube, and Main; the Oder flood in July/August 1997; the flood in Elbe and Danube in August 2002; the floods in Austria and Switzerland in August 2005. Less mentioned in international literature is the fact that this weather is responsible of extensive floods events and of triggering landslides, also in the southern sector of the Alps and in
Norway. Roald (2008) documented many flood events caused by Vb cyclones in southeastern Norway, like the flood in July 1789, the one in 1860 and the more recently the flood in June 2011, among others.

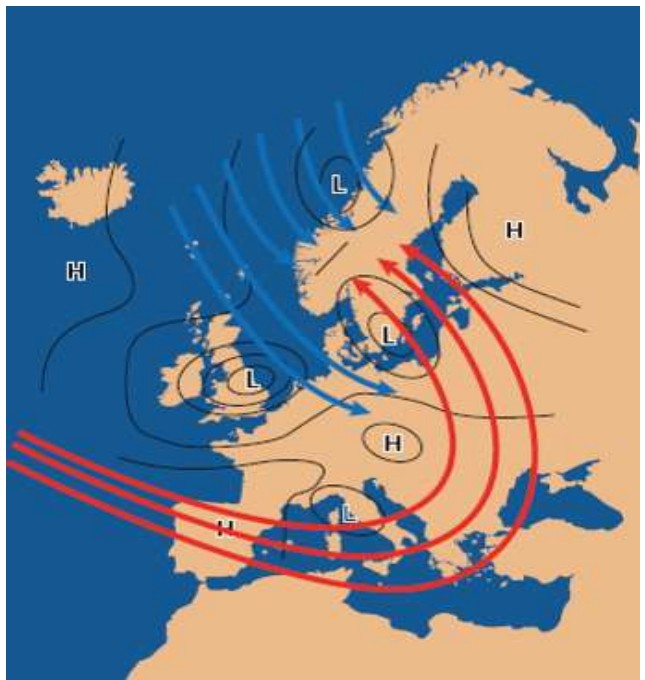

**Fig. 7:** Southern type of weather: Vb trajectory. Where "H" is high-pressure and "L" is low-pressure. Red lines indicate warm and humid air, while blue lines represent cold air masses (source, NVE)

# 5 Meteorological conditions in late spring 2013

The meteorological pattern that affected Europe on late spring 2013 started at the end of April 2013 and lasted until the beginnings of June 2013. In Piemonte major impacts were observed in the first half of this period, while in Norway from 15th May to 2nd June 2013.

## 5.1 Antecedent conditions

The winter 2012/2013 was relatively cold (-0.23 °C respect the 1977-2001 average temperature) and dry (-52% respect 1977-2001 average precipitation) in both countries with temperature lower than normal while it was still cold in Piemonte from March to May, but wetter than in Norway. On March, rainfalls in Piemonte were +30% above normal and heavy precipitations happened on the end of April. Between 27th April and 1st May 2013 several rain gauges in northwestern Piemonte recorded more than 250 mm in five days (against 175 mm average April precipitation).

In southeastern Norway, the period January-April 2013 was cold and dry than normal and characterized by cool air from North (Roald, 2015). The average temperature was 2-3° below the normal in this area especially in the interior northwestern part close to the mountains. The spring arrived late. The period January-April was characterized by precipitation deficit in many areas. The precipitation was 90% of the normal in the entire country, however southeastern Norway received only 25% to

50%. The snow depth was lower than normal and thus ground frost deeper than normal. In May, the warm air from south and south-east initiated snow melt in the mountains. **Fig. 8** shows the snow distribution in Norway during April-May 2013. On middle May, there was still snow cover above 700m asl and more snow depth than normal in the western part of the area. On May precipitations were +200-500% above average, especially in the western parts of the area.

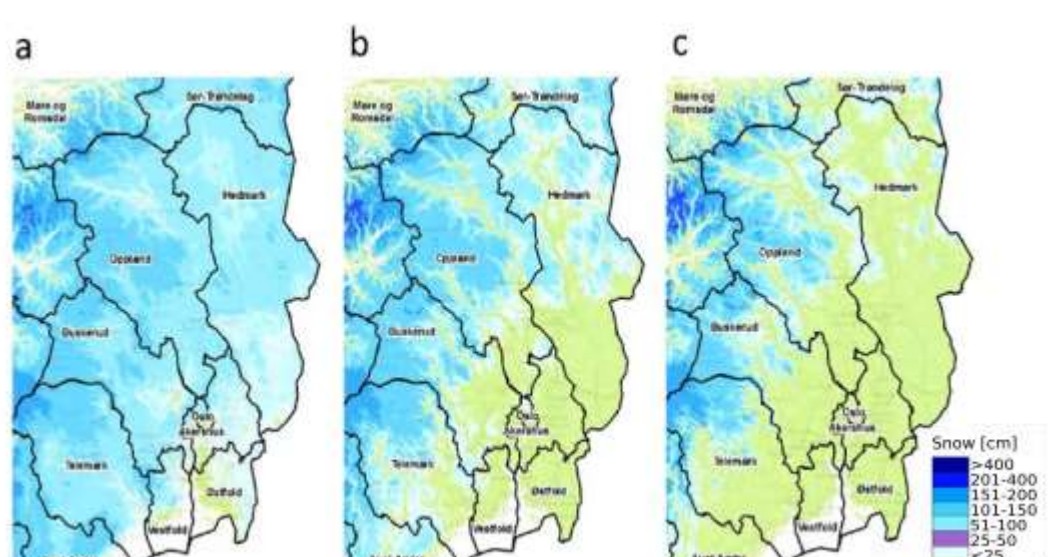

**Fig. 8:** Snow distribution a) middle of April, b) end of April and c) middle May, Norway (source: xgeo.no)

## 5.2 Meteorological conditions during the period analysed

The Vb atmospheric pattern influenced weather in Europe, and in particular in Italy and Norway, for a long time during Spring
10 2013, from the end of April until the beginnings of June 2013. In retrospect, it could be observed that the Vb weather regime was relatively easy to follow across the Mediterranean Sea. In the initial phase, the Vb cyclone was responsible of strong winds (up to 120 km/h) that produced sandstorms at Malta and southern Italy (particularly in Sicily and Calabria region), the 15[th] and 16[th] May 2013 with some impacts on population (Meteoweb, 2013). While the system moved toward north, it was responsible of producing intense rainfall in proximity of the western Alps. The Vb system continued to northern Europe bringing warm
15 air at higher latitudes and rainfalls when it reached in southern Norway.

### 5.2.1 Piemonte

From 15[th] May 2013 to 19[th] May 2013, an intense cold front affected Piemonte, causing abundant precipitations, a general increase of rivers discharge and vast areas of Piemonte were affected by floods and landslides.

On 15th May 2013, a trough blunder on Western Europe conveyed warm and wet flows from south towards Piemonte, causing widespread precipitations that intensified especially in the northern Piemonte and on the border areas with Liguria. **Fig. 9** shows the mean sea level pressure analysis by the global numerical weather prediction model operated by the European Centre for Medium-range Weather Forecast (ECMWF) on 16th May 2013 00:00 UTC. The main low-pressure system is centered over the North Sea, while a secondary low one is near North Africa coasts: isobars determine intense southern humid air-flow from Mediterranean Sea towards Scandinavian Peninsula.

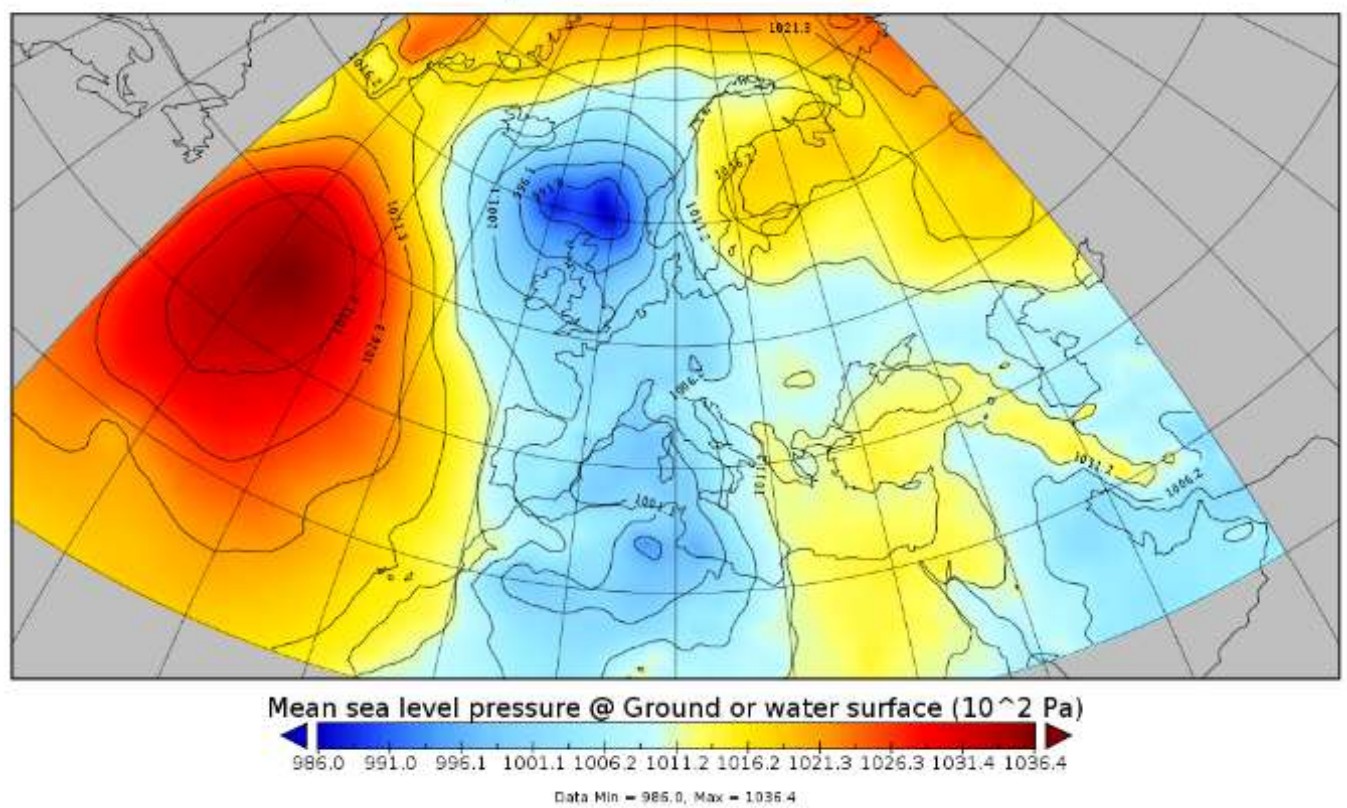

**Fig. 9:** ECMWF mean sea level pressure analysis over Europe on 16th May 2013 at 00:00 UTC (source: Arpa Piemonte).

Widespread moderate, locally strong, precipitations affected Piemonte during night. In the Po valley were recorded on average 30 - 40 mm rainfall with 45.6 mm maximum over 24 hours. About 20-25 cm fresh snow were recorded in the Alps above 2000 m asl. On 16th May 2013, the low-pressure area, responsible for severe weather, gradually moved towards Biscay Bay, continuing to convey wet and unstable air over Piemonte. However, an increase of atmospheric pressure in Ligurian Gulf caused an attenuation of meridional flows and a general attenuation of precipitations. In the late evening, the cold front

associated with the low-pressure crossed Piemonte, causing instability and convective rainfall, more intense over the north-western footpath. On 17[th] May afternoon, the cold sector, which affected Piemonte over past 48 hours passed, favoring a general attenuation in rainfalls. However, atmospheric post-frontal instability caused sparse thunderstorms, particularly on the western foothills, where the interactions between southern flows and alpine foothills caused strong connection with abundant hail: hourly precipitation rates reached even 40 mm. Discharges of minor hydrological network increased as result of severe thunderstorms. On 18[th] May 2013, the occluded front passed Piemonte from west to east. The wet airflow remained intense from south, resulting in convergence close to the northwestern foothills, with further intensification of rainfalls. On 19[th] May 2013 morning, the sea level pressure increased, favoring the precipitation exhaustion, except for northern Piemonte where rainfall terminated during central hours. The rainfalls between 18[th] and 19[th] resulted in significant increases in rivers discharge both in northern and southern basins. Attention levels of were reported on secondary hydrological network, particularly in the basins near Turin. Over the entire period, more than 300 mm fell in northwestern areas with peaks of 350 mm in 96 hours (**Fig. 10**). The return period for 3-6 hours rainfall accumulation were about 20 years. Finally, several catchments recorded more than 600 mm since 1[st] March 2013 to mid-May.

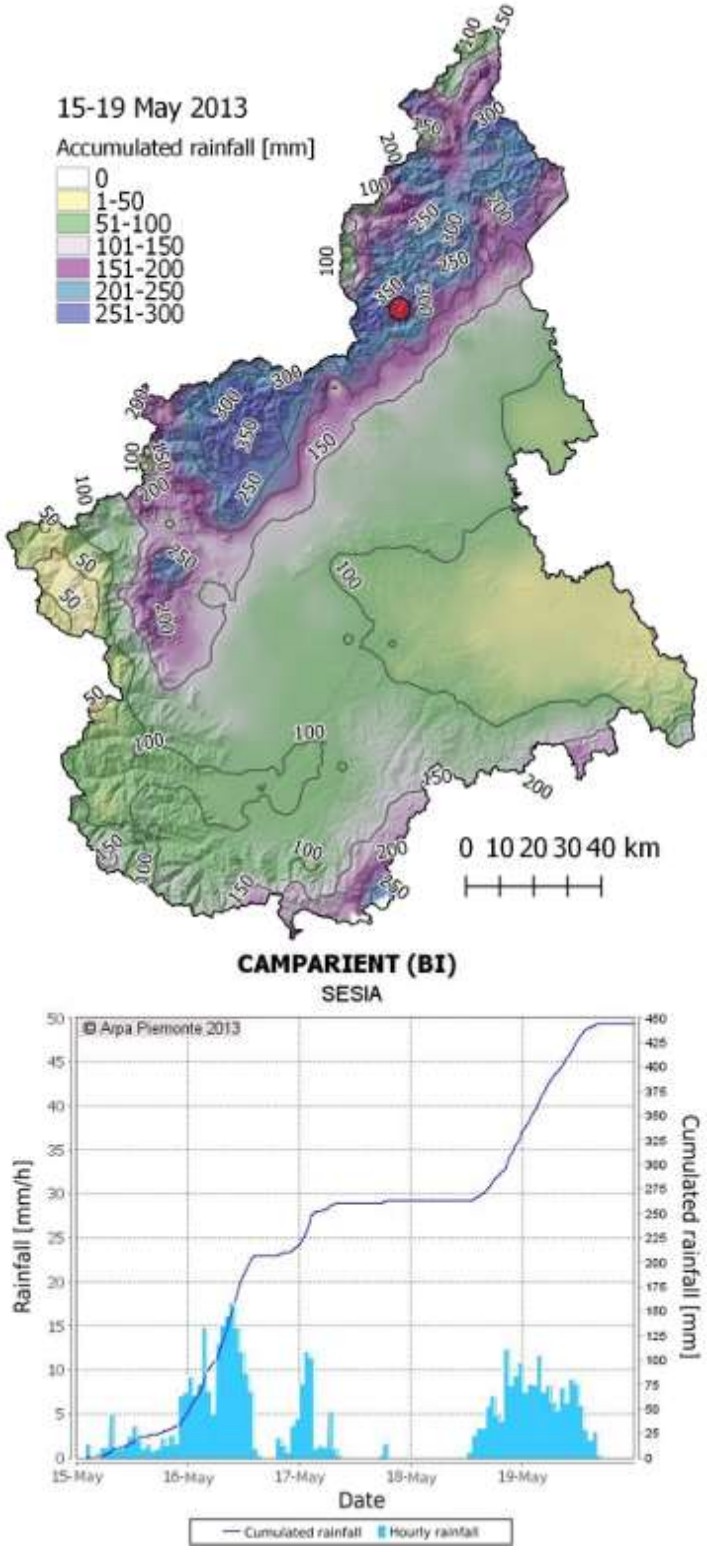

CAMPARIENT (BI)
SESIA

© Arpa Piemonte 2013

**Fig. 10:** Accumulated rainfall from 15th May to 19th May 2013 (source: Arpa Piemonte). As example of rainfall distribution during the event, the ietogram recorded by Camparient rain gauge (red dot in the accumulated rainfall map) is showed.

### 5.2.2 Norway

It is well known, in Norway, that Vb cyclones can produce the largest floods during spring (Roald, 2008). Therefore, the flood forecasting service at NVE, pays attention, each year, to these weather conditions in southern Europe. The arrival of the Vb cyclone on May 2013 was forecasted with some weeks in advance by mentioning the possible arrival of a warm weather system from south (as it was indicated in a "situation report" published the 3rd May). The Vb cyclone brought warm winds with high temperatures that caused intense snow melt over a large area. The temperature starts to increase around the 5th May in the mountain area, starting the snow melt process. A short decrease in temperature was observed on 14th -15th May and 22nd May before and in correspondence of the two main rainfall episodes. Temperatures reached the highest peak on 18th and 19th May, causing significant snow melting in the area. Due to the arrival of several warm air fronts temperatures continued to increase constantly until the end of May.

The first rainfall arrived on 15th -16th May and in the easterly counties of Telemark and Buskerud (**Fig. 11a**). In Eggedal station were measured 60 mm in 24 hours. In this area, many hydrologic stations reach the flood level (**Fig. 11b**) and in Eggedal the water discharge was the fourth highest since registration started in 1972, resulting in a big flood.

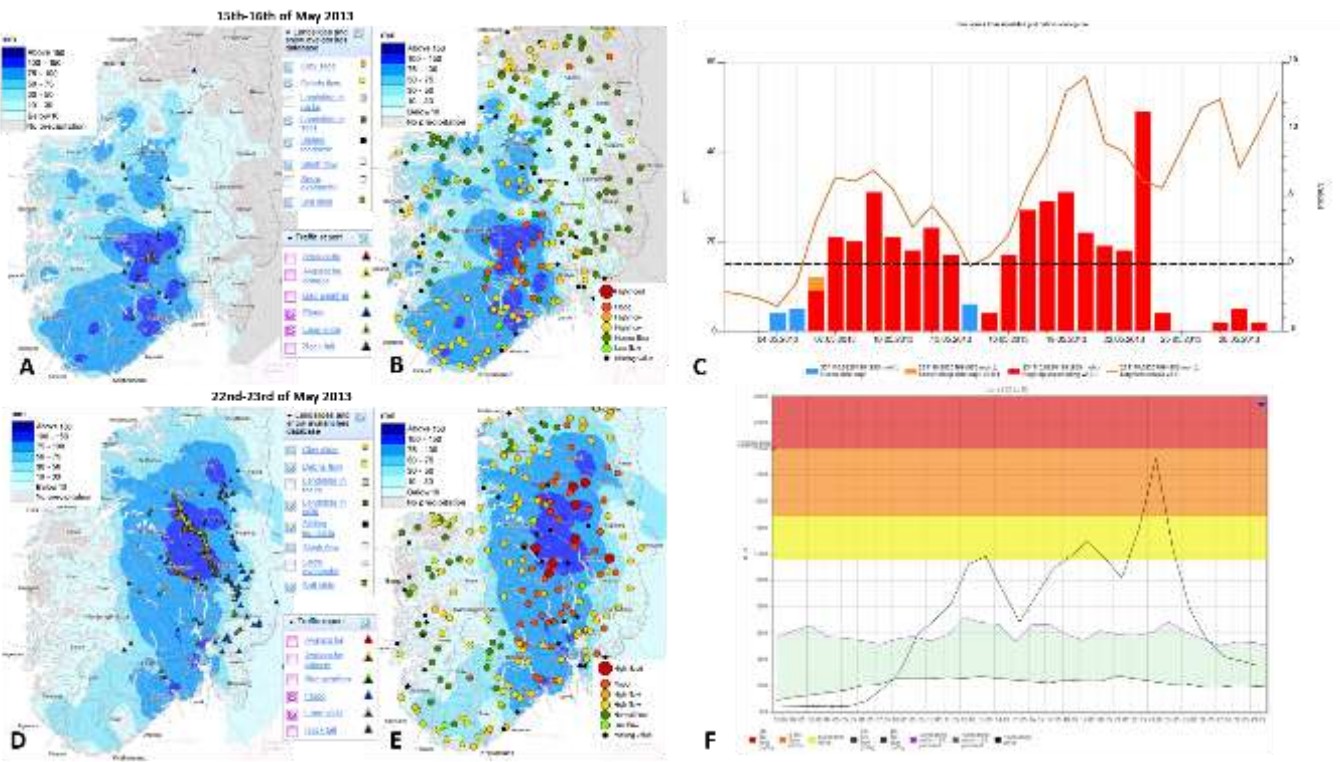

**Fig. 11:** The 15th-16th May and the 22nd May events. A) Rainfall measurement the 15th and 16th May and landslide events; B) Rainfall measurement the 15th and 16th May and water discharge level; C) Rainfall and temperature distribution during May in the Gudbrandsdalen area (red: rainfall; blue: new snow, orange line: temperature); D) Rainfall measurement the 22nd and 23rd May and landslide events; E) Rainfall measurement the 22nd and 23rd May and water discharge level; F) Water discharge

at Etna station during May 2013 (source: xgeo.no).

Precipitations started in the western counties, moving eastward. A second and more significant rainfall episode occurred on the 22nd and 23rd May affecting mainly the Glomma and Østerdalen catchments in the eastern sector of the region (**Fig. 11d**). This initiated a large flood along Glomma river. In Østerdalen were measured 50-60 mm that day, while in Gudbrandsdalen

values ranged from 50 mm to 93 mm. An overview of the rainfall and temperature distribution during May 2013 in the Gudbrandsdalen area is presented in **Fig. 11c**. The two rainfall episodes, in addition to the incoming snowmelt, were responsible for the increase of groundwater in the region and to produce high water discharge in many of the rivers in the area (**Fig. 11e**). In Folla river a flood of 100-year return period was observed, while in Numedalslågen and Skien catchments a 30-year return period flood was observed (Roald, 2015). In Drammen river at Begna, Etna and Dokka station the flood reaches

the 50-year return period (**Fig. 11f**). At Gausdal and Gudbrandsdal river the flood was estimated to be between 50 and 100-year return period. The Vb situation persisted from 15th May – 2nd June and caused also intense rainfall and urban flood in the capital Oslo on the 2nd of June.

## 5.3 Warning levels

The warning levels indicate landslide hazards and generally the actions that a recipient should undertake to reduce potential

damages. Both services at ARPA Piemonte and NVE use similar levels symbolized by the typical traffic lights colors, summarized in the **Table 2**. Even if the numbering of the levels is different, the meaning of the warning is similar. Emergency response authorities should be prepared to implement emergency plans, mitigation measures, carry out evacuations and other contingency responses. Hazard and risk maps are mandatory to help local authorities to prioritize the implementation of measures.

**Table 2** – Warning levels in use in Italy and Norway, with their respective local names.

| Warning level | Italy | Norway | Explanation |
|---|---|---|---|
| **Red** | 3 | 4 | Very high landslide hazard. Many landslides and several of large dimensions may occur; their long runout and extent may result in damage to settlements and infrastructure. Red level is an extreme situation that occurs very rarely. Emergency response authorities should have implemented emergency plans, appropriated safety measures, such as closed roads, and carry out evacuations. Follow authorities' recommendations. |

| Orange | 2 | 3 | High landslide hazard. Many landslides and some of large dimensions are expected. Incidents that can impact infrastructure and roads may occur. Emergency response authorities should increase vigilance, and should be prepare to implement emergency plans, evaluating the needs for evacuation and carrying on safety measures. Exposed roads may be closed off. Pay attention to media and follow recommendations from the authorities. |
|--------|---|---|---|
| Yellow | 1 | 2 | Moderate landslide hazard, primarily small slides may occur, on artificial slopes that may affect roads, railways or along river embankments. Sparse debris avalanches or debris flows (also of large dimensions) may also occur causing damage to infrastructure and people, but primarily on a local scale. Emergency authorities should increase vigilance and pay attention to weather conditions and landslide forecasts. Preventive measures are recommended. |
| Green | 0 | 1 | Generally safe conditions. Debris avalanches, debris flows, shallow slides and slushflows are not expected at this level. |

In Norway, warning levels are updated two times a day. The warning messages are sent from 66 h to a few hours ahead.

On 15th May 2013 the first warnings in Northwestern Alps and in central hilly areas about possible debris flows and landslides were issued by ARPA Piemonte (**Fig. 12**). Then, according to observed precipitations and updated NWP outputs, the warning levels remained stable over Alps, while alerted hilly areas reduced. According to observed rainfall occurred during 18th May and the first hours on 19th May 2013, warning for local floods, debris flows and landslides was newly issued for central Piemonte.

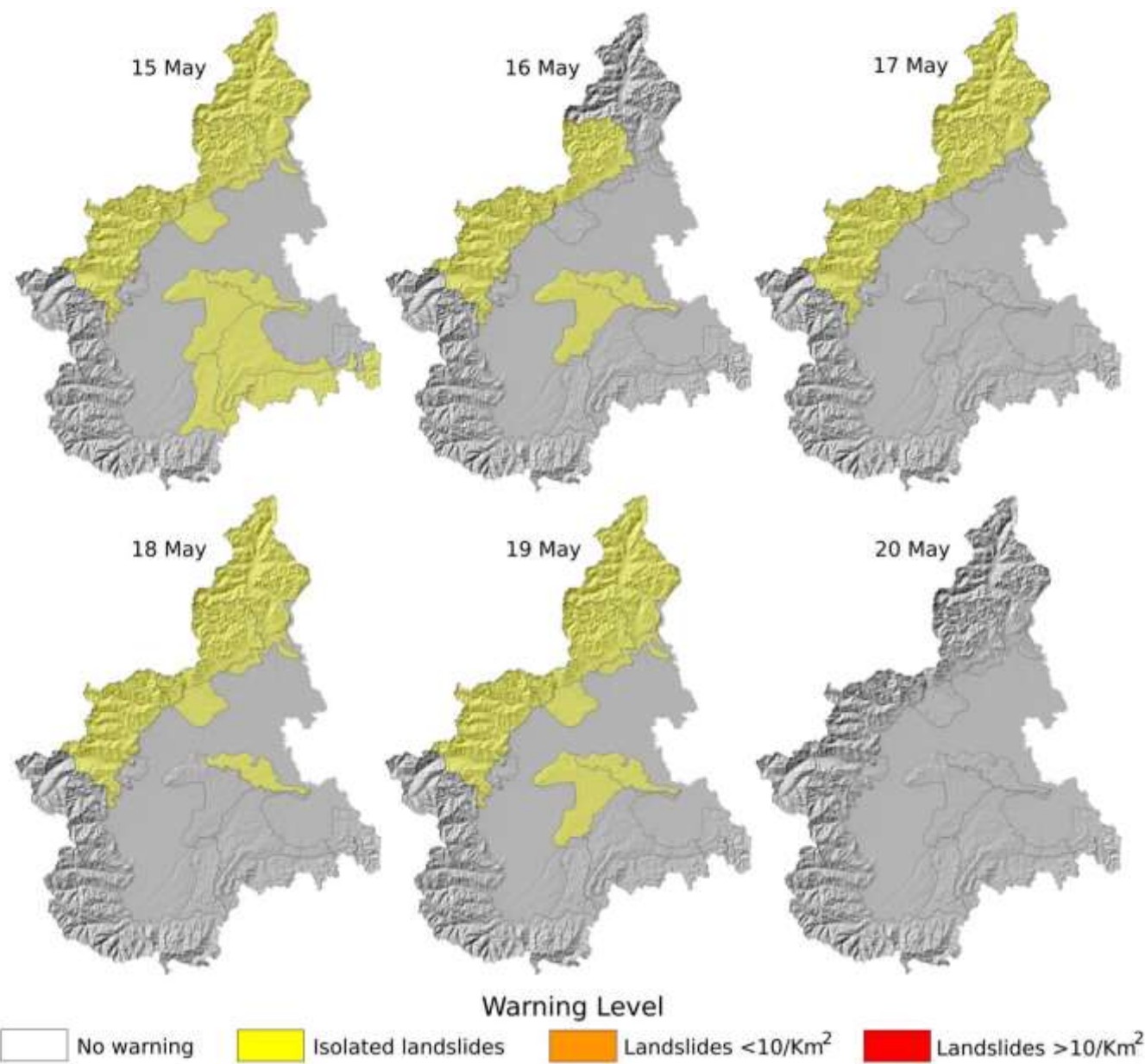

**Warning Level**

No warning · Isolated landslides · Landslides <10/Km$^2$ · Landslides >10/Km$^2$

**Fig. 12:** Landslide warnings issued from the 15$^{th}$ to the 20$^{th}$ May 2013 in Piemonte region (source: Arpa Piemonte).

A first flood warning was sent in Norway the 14$^{th}$ May 2013, followed by a first landslide warning (orange level) on 15$^{th}$ May,
for parts of the southeastern region. A yellow level was kept from 16$^{th}$ to 20$^{th}$ May. On 21$^{st}$ May the level was increase to
orange and on 22$^{nd}$ May was also added a red warning level (**Fig. 13**). Different landslide warnings were issued every day until

the end of May as shown in the **Fig. 13**. From 16th May until the end of May a yellow warning was also sent for Northern Norway.

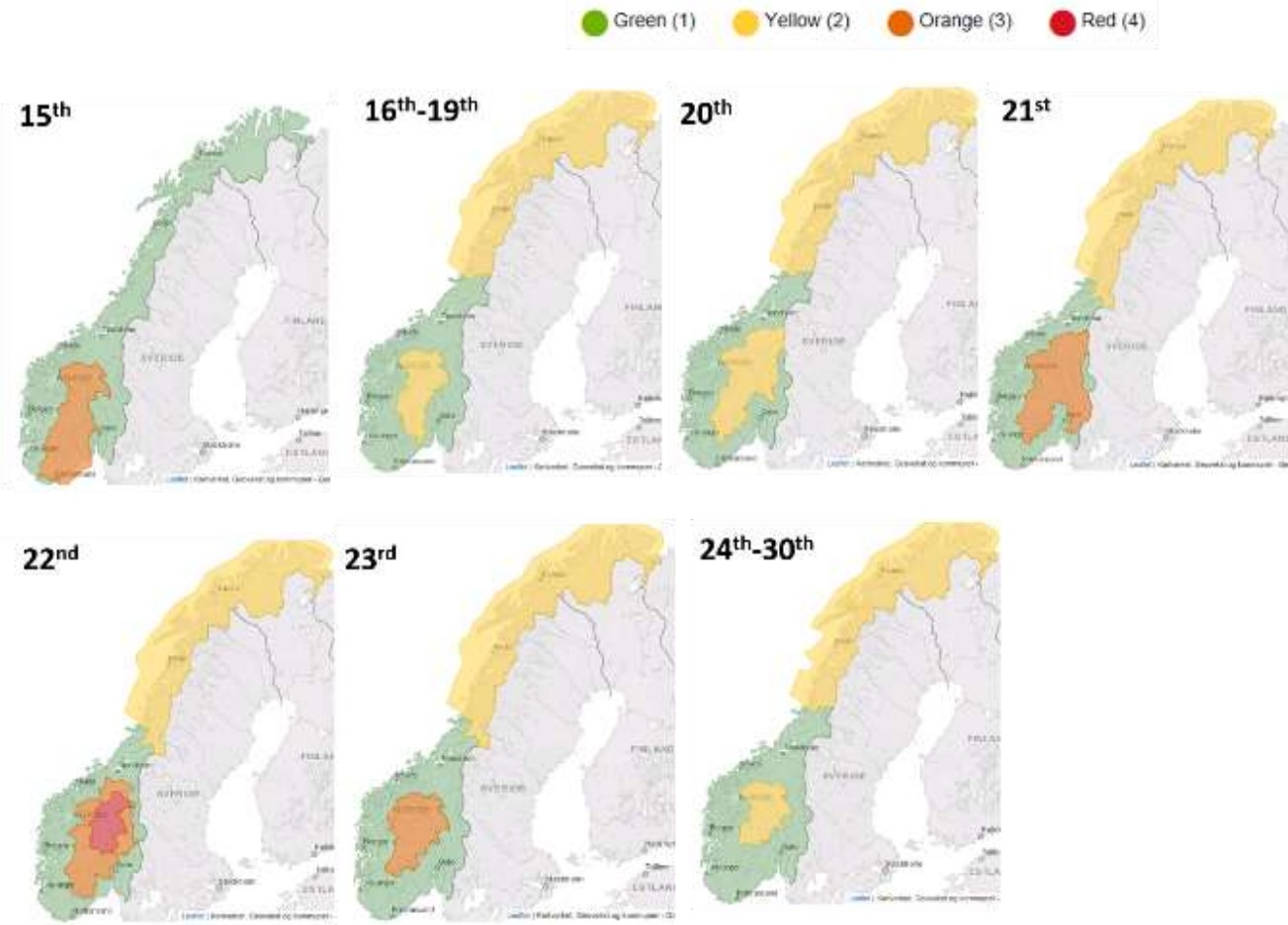

**Fig. 13:** Landslide warnings issued from the 15th to the 30th May 2013 in Norway (source: varsom.no).

The severity of the rainfall and snowmelt episode that occurred on 22nd May was clearly detected with some days in advance when the Hydmet map shows that landslide thresholds would be exceeded (**Fig. 14**).

## Prognosis of landslide thresholds

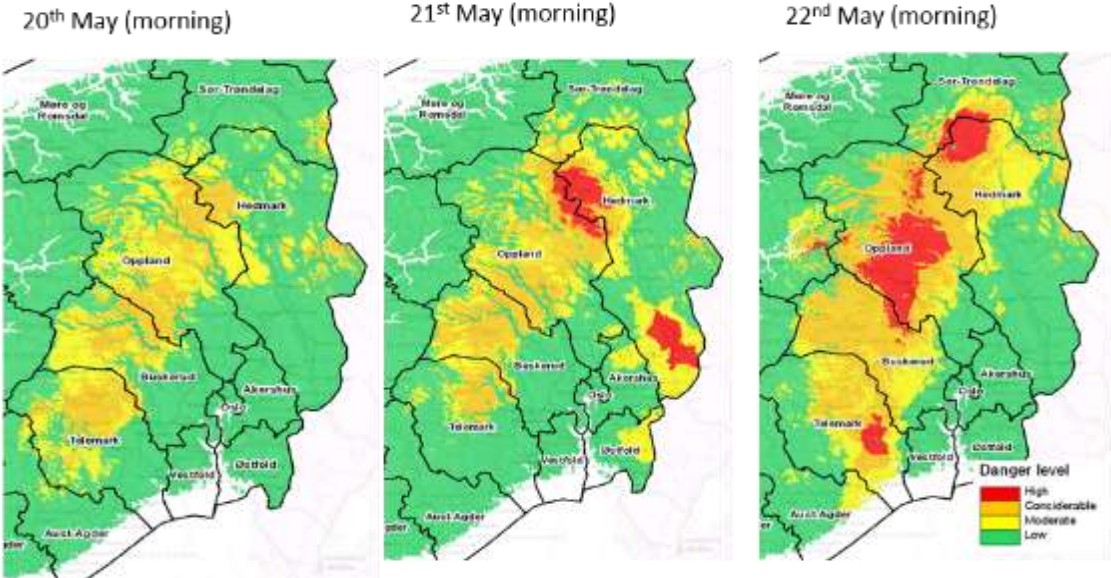

**Fig. 14:** Prognosis of landslide thresholds (source: xgeo.no).

# 6 Type of landslides and economic consequences

In Piemonte, after these rainfall events, about 320 slope phenomena have been reported (300 landslides and 20 channelized debris flows) (**Fig. 15**). The main landslide types occurred were wide shallow landslides, deep-seated rotational slides in alpine and hilly areas, and subordinated reactivation of some translational slides in hilly environment. The territory hit by slope and flood processes (**Fig. 16**) covers an area of 3,700 km$^2$ with about 420,000 inhabitants. In affected areas, important connecting routes are sited, including international ones. Due to severe phenomena occurred, numerous cases of traffic interruptions by landslides and flood were reported, as well as flooding of buildings, deposition of coarse alluvial sediments on roads, bridges jammed by debris flows, roadside walls collapse, erosion of roads surface, urban flooding and people trapped in cars. Rainfall event caused wide discomforts and damages to the community, both in relation to normal social cohabitation and economic wealth.

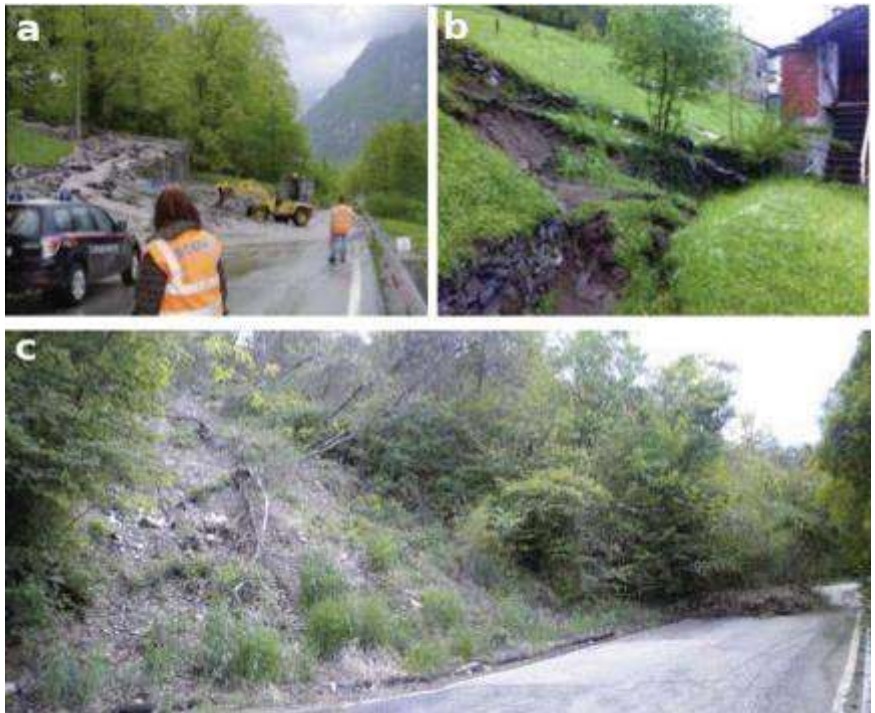

**Fig. 15:** Some examples of slope phenomena recorded during rainfall event. a) Road interrupted by a debris flow; b) a building hit by small rotational slide; c) shallow landslides on a road (source: Regione Piemonte).

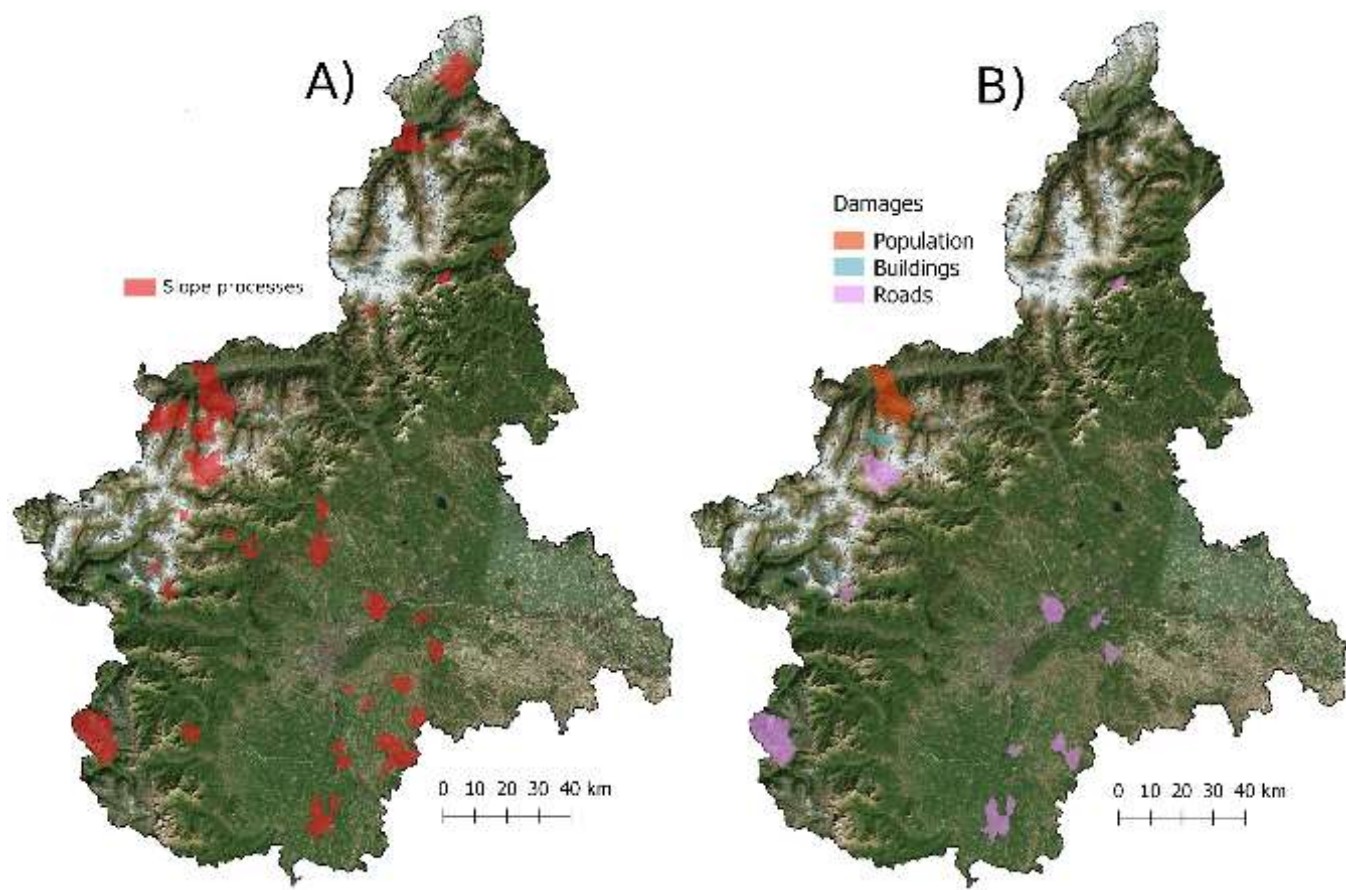

**Fig. 16:** A) Landslide processes occurred during the rainfall event. B) Reported landslide damages (source: Arpa Piemonte).

**Fig. 17** shows the warning areas issued (in yellow) and the municipalities where landslides were reported (in red): the derived contingency table gives a probability of detection (POD) equal to 0.70 (benchmark equal to 1) and a false alarm ratio (FAR) equal to 0.63 (benchmark equal to 0).

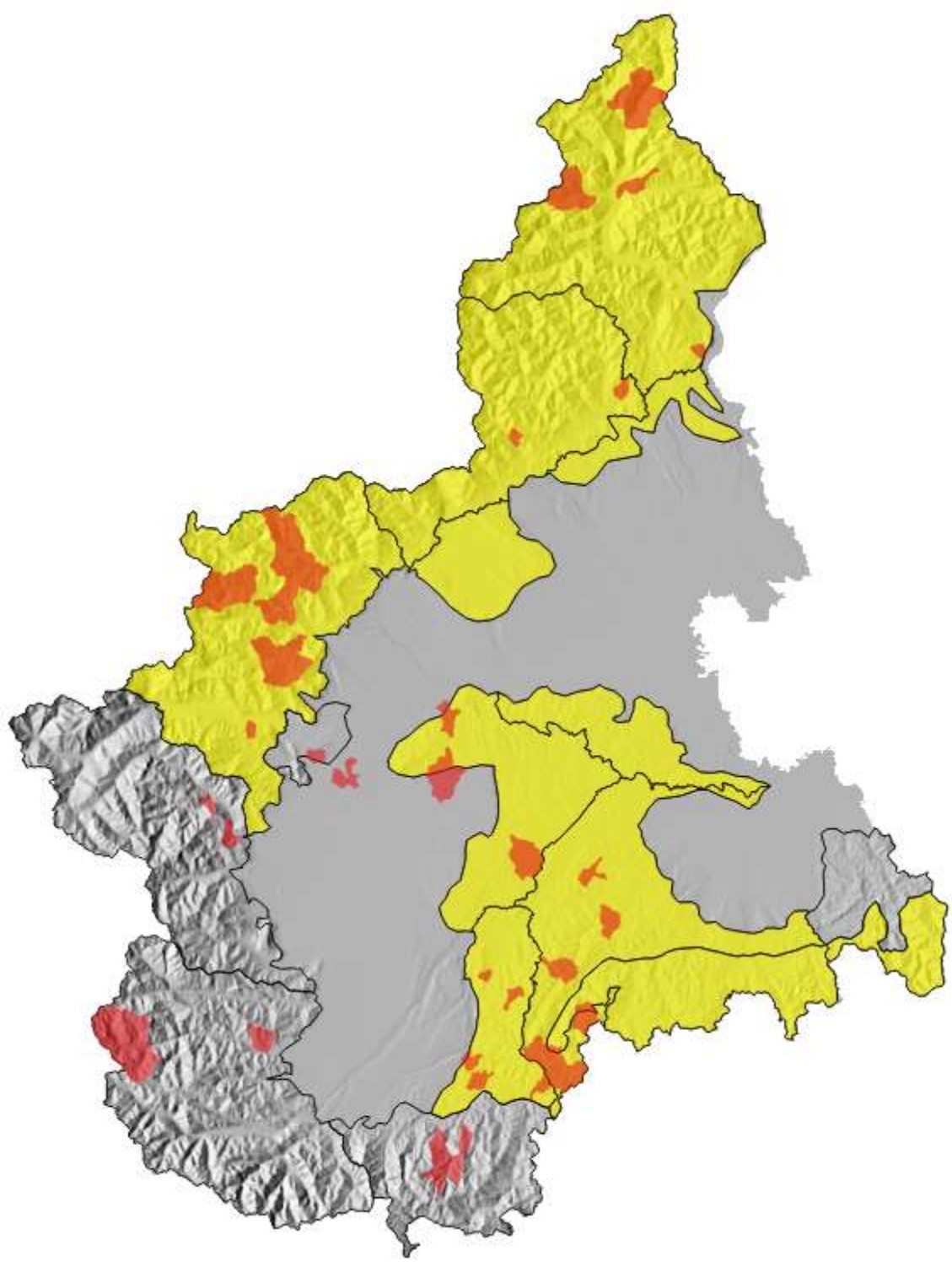

**Fig. 17:** Warning issued (yellow polygons) and observed wide landslides area (red polygons) during the event.

In Norway, more than 100 landslides were recorded in the database ([www.skredregistrering.no](www.skredregistrering.no)) in this region between 15[th] May and 7[th] June. Mainly the events that reached roads and railway were recorded, but we believe that many more have occurred, but they have not been reported due to their location in less inhabited areas. **Fig. 11a** shows the spatial distribution of landslides during the first rainfall event and **Fig. 11d** shows the landslides occurred during the second rainfall event. In both figures are also indicated the areas where the road was blocked because of flood. The landslides observed were mainly debris flows and a combination of debris slides and debris flows, however most of them have been reported generically as landslides in soil (**Fig. 18**). There were many shallow slides in artificial cuts, mainly translation of small dimensions or along river sides. A few slushflows were also reported, especially in the northern sector of the region in the mountains. The landslides events and floods produced significant damages to roads, railway and private buildings and the economic losses were estimated around ~170 MI € (~1.5 bill NOK) for 22[nd] May. Many places were evacuated for several days. There were 350 cases of damage and 23 municipalities asked for mitigation measures in Hedmark and Oppland. The same system triggered landslides and floods in Northern of Norway, but this is outside the scope of this paper.

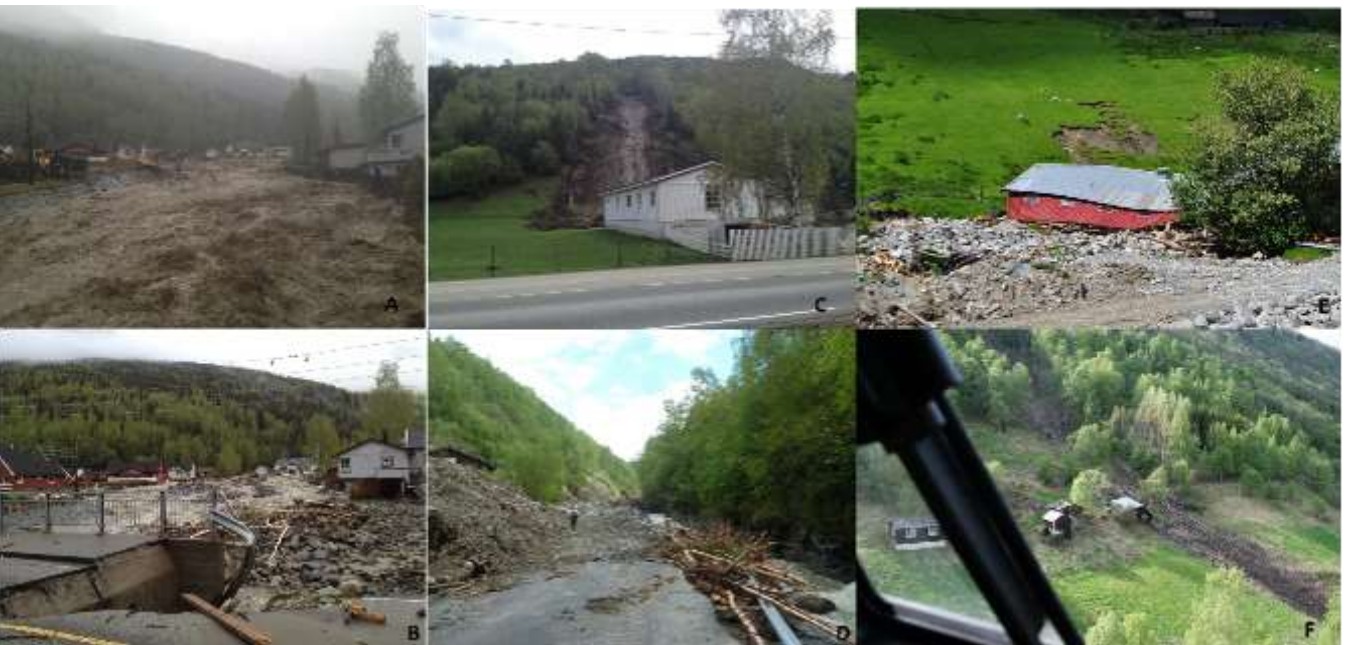

**Fig. 18**: Flood and landslides at Kvam, Nord Frøn, Oppland county 22[nd] May 2013. A) and B) Flood and flood damages at Kvam; C) Debris slide at Ringebu; D) debris flow at Veikleåadalen E) shallow debris slide and debris flow deposits at Kvam; F) Debris flow at Veikleaådalen (Source: NVE).

# 7 Discussion and conclusions

Two territorial landslide forecasting and warning services operating in two European countries, Italy and Norway, have been presented and compared. They were designed to predict rainfall- and snowmelt induced landslides, general term used to refer to rapid mass movements, like shallow translational and rotational slides, channelized debris flows, debris avalanches. The Norwegian system is also able to predict slushflows. The organization of both services started at governmental level in the late

2000's as part and in synergy with flood and snow avalanches forecast services. Using statistical methods landslide thresholds were derived. Rainfall ID thresholds are used in Piemonte defined for the different types of landslides, while in Norway a unique threshold based on water supply (rain and snowmelt) and soil moisture is used for all type of landslides and for the entire country. However, regional adaptation of the thresholds are in progress. Landslide thresholds can be visualized in form of maps in their respective web interfaces and expert tools. Daily landslide hazard assessments are made on the basis of expert

knowledge combined with quantitative thresholds, regular rainfall forecasts and real-time observations. Both services use four warning levels, that are published and disseminated through internet. Information from news, field survey are used to verify the landslide occurrence. The evaluation of the performance of the systems is under development.

In this study, we presented a case of successful forecasted landslide events, triggered by the same large atmospheric synoptic pattern, the Vb cyclone, that occurred across Europe on May 2013. Even if Vb cyclones are rare phenomena, and probably

expecting to decrease in frequency in the future (Messmer et al., 2015), they are serious challenges for forecasting services and emergency response authorities during spring, because they are large atmospheric patterns that can affect areas for a long time and they produce large floods and hundreds of landslides. The Vb cyclone on 2013 triggered flood and hundreds of landslides, mainly debris flows, debris slides and shallow slides both in Piemonte, Northwestern Italy, and in Østlandet region, Southeastern Norway, producing severe damages to infrastructure and buildings. The type of landslides triggered by the

described event were similar in both countries, like debris flows, debris slides and relatively shallow slides, not particularly large in term of volume, but in large amount and largely spread all over the regions, causing significant damages to infrastructure or isolating communities because of blockage of roads and railways. Even if Piemonte and Norway forecasting services worked separately, and they differ in some components (like landslide thresholds) they emitted accurate warning messages at regional level that were extremely useful for road and railway administrations and municipalities. Based on these

warnings, stakeholders activated timely emergency plans, mitigation measures, evacuations and other contingency responses before the events. It is important that the different forecasting services follow the system since the initial stage to be better prepared. As presented herein, the predictability of Vb cyclones is quite good, however, they are very large synoptic systems and damages cannot be completely reduced, therefore society and operational services must be prepared. For forecasting services, this condition is also demanding, because require a lot of personal on duty at the same time for both flood and

landslide forecasting.

This study is the result of an inter-institutional and international collaboration across Europe, initiated and promoted in 2016 by the Norwegian Water Resources and Energy Directorate (NVE) in two ways:

a) creating a network of experts working in the prevention of rainfall-induced landslides that were gathered together during an international workshop in Oslo in October 2016 to establish a forum for exchange of knowledge, challenges

and best practices among those working with operational forecasting services for rainfall-induced landslides (Devoli, 2017);

b) promoting collaborations with specific institutions to study specific events: besides the one described herein with ARPA Piemonte, NVE is collaborating with the British Geological Survey to compare forecasting experiences and to better understand "westerly" synoptic systems, that move across the Atlantic in the autumn, causing landslides in UK and in Norway (like the storm Desmond, the 4[th] and 5[th] December 2015).

This study demonstrates the good skill and usefulness of shallow landslides EWSs in case of large synoptic forcing like Vb cyclones in different countries. Moreover, it demonstrates that international collaborative efforts among natural hazards prediction centres operating in different countries can be very useful, because they can improve knowledge in natural hazards associated to these large synoptic systems increasing the lead time and the forecasting effectiveness of the warning services.

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
