# Peer review of "Comparison of landslide forecasting services in Piemonte (Italy) and in Norway, illustrated by events in late spring 2013"

_Natural Hazards and Earth System Sciences, 2017_

## Referee Comment (RC1) · Anonymous Referee #1 · 21 Dec 2017

**RE:** NHESS 2017 411          Tiranti et al.  Regional landslide forecasting in Piemonte (Italy) and in Norway: experiences from 2013 late spring

**Overview**

Two different warning systems against rapid landslide phenomena hazard are shown in present work. The work is, in general, well written but has two main deficiencies:

1) Some details about phenomena and warning thresholds are missing
2) A final comparison between warned phenomena and those occurred is missing

Landslide phenomena "warned" by the two systems seem different.

In the case of ARPA Piemonte shallow landslide, deep landslides and channelized debris flows. ARPA Piemonte has three different forecasting approaches: one for each type of phenomenon.

In the case of Norvegian Forecasting Service, the "warned" phenomena are: shallow landslide, debris avalanches, slushflows and debris flows. In this case, the writer supposes that debris flow is landslide induced debris flow. Is it right?

Then, the writer suggests a section after the introduction where all the "warned" phenomena are briefly introduced and schematized. This section would help the reader to focuse the phenomena, increasing the value of the work.

About Norvegian Forecasting Service, there is an unique threshold for all the type of landslides. Authors should justify this.

At the end a comparison between launched warnings and occurred phenomena should very useful for understanding the effectiveness of the warning systems. As in the case above, this should increase  the value of the work and consequently its diffusion.

The following are the detailed comments and specifications.

**Abstract**

Sentences at lines 14-17  ("In Italy,................in Northwestern Italy.") are redundant. Just write that in Italy the landslide hazard assessment is not national but provided by the Regional Agency for Environmental Protection and that in present paper it is shown the work of ARPA Piemonte.

Sentence at lines 24-27 is too long.

**Introduction**

Sentences at lines 7-16 ("The spatial occurrence………shallow landslides") appears confuse. At the beginning rainfall and snowmelt induced landslide are introduced. After that rapid mass movements are introduced and rainfall and snowmelt induced landslide becomes a subcategory of them. Finally, there are the shallow landslides. Authors should to identify all the phenomena they warn to avoid confusion in the reader.
At line 19, you could insert the reference Thiene et al., (2016).

**The Vb cyclones**

About line 14 of page 3, are you sure about" eastern part of the Alps to NorthWest?"; maybe it could be the contrary.

**The landslide forecasting services in …**

At line 14 of page 9: "is not based only on a threshold" but also on………

**Piemonte's landslide forecasting service**

About the use of DEFENSE. What type of weather radar is used? In some cases after using movable weather radar it could be possible reliably estimate the rainfall depth. In a mountain environment in many cases, radar estimates could be not reliable due to the high spatial variability of rainfall and the distance from the fixed weather radar (Germann et al., 2006; Rabiei and Haberlandt, 2015; Bernard et al., 2016).
Authors should introduce some details and cautions on the use of weather radar in a mountain environment.

**The Norwegian Forecasting Service**

This section is a bit confused and should be reorganized and rewritten. Initially, it is stated that the daily assessment is built on thresholds, real time observations and landslide events (occurring

during the event?) previous occurred landslides (inventory) and susceptibility maps. It is reasonable that the inventory of landslide be used for determining the threshold. Is not it?

Moreover, also runoff is simulated (by which model?): which is the scope if the threshold combine rainfall and snow with the soil saturation degree?

In other words, it is better only describe what it is strictly related to the landslide warning.

At last most references are not in English language. Therefore, some detail about model could be written in an appendix.

**Antecedent condition**

At line 6 of page 13, it is better 5 days instead of 120 hours.

**Piemonte**

What is ECMWF at line 12 of page 14?

In the sentences of pages 15-16 a big emphasis is addressed to the occurred flood and the reaching of warning level in the hydrological network. The writer think that this is eccentric respect to the main object of this work, the landslide forecasting. Therefore, the writer suggests the authors to give a brief description of occurred flood and stress the description of occurred phenomena related to the landslide hazard object of forecasting.

**Warning levels**

The first sentence states that to each warning level corresponds a measure. Is it true? In this case further details will be useful.

About debris flows warned (see line 5 of page 20) which radar was used?

Bernard M., Stancanelli L., Berti M., Simoni A., Gregoretti C., Lanzoni S. (2016) *Field results from the runoff generated debris flows occurred at Rovina di Cancia (Venetian Dolomites)* XXXV Convegno di Idraulica e Costruzioni Idrauliche – Bologna

Germann, U., Galli, G., Boscacci, M., and Bolliger, M. (2006). Radar precipitation measurement in a mountainous region. Quarterly Journal of the Royal Meteorological Society, 132(618), 1669-1692. http://doi.org/10.1256/qj.05.190

Rabiei, E., and Haberlandt, U. (2015). Applying bias correction for merging rain gauge and radar data. Journal of Hydrology, 522, 544-557. http://doi.org/10.1016/j.jhydrol.2015.01.020

Thiene, M., Shaw, W. D., and Scarpa R., (2016). Perceived risks of mountain landslides in Italy: Stated choices for subjective risk reductions. Landslide, doi:10.1007/s10346-016-0741-3.

---

## Referee Comment (RC2) · Anonymous Referee #2 · 2 Jan 2018

I have read and carefully evaluated the manuscript "Regional landslide forecasting in Piemonte (Italy) and in Norway: experiences from 2013 late spring", submitted to NHESS by Tiranti et al.

The manuscript describes a large cyclone system that struck two very different and distant test sites (Norway and northern Italy). Plenty of details are provided about the event, the related hazard and how it was managed in the two sites.

The topic is very interesting and completely centered on the scopes of the journal and of the special issue.

However, what we have here is more a couple of (interesting) event reports than a

"real" research paper. Since NHESS does not consider manuscript types such as reports, technical notes, event descriptions and so on, my recommendation is to perform "MAJOR REVISIONS" in order to better highlight and discuss the interesting scientific and research outcomes that I saw in the manuscript. I want to stress that I consider this manuscript a very interesting contribution and that it has a very good potential. It just needs to be put in the right perspective (research-oriented paper instead than an event report). My subsequent comments are aimed to this change of perspective. I encourage the authors to carry on and adjust the manuscript to make it become a high-quality research article.

GENERAL COMMENTS

1 The English is fully understandable and good (although I identified some minor errors). I am not a native speaker, however I suggest to check the text again and especially to split long sentences. They may be hard to follow.

2 The manuscript is very rich of details. At the beginning I was positively impressed, but after several pages it got quite boring: the description of the event and of the EWS infrastructures are very good but sometimes the scientific content and the research aspect were missing. And I think those are the main things that NHESS readers are expecting form a research paper. I suggest reducing the description (maybe using a few tables?) and to let research topics stand out from the text (see e.g. my following comments).

3 Another issue is that the two cases of study stay always kept separated. There must be a point in the manuscript when things are put together and a synthesis is made to highlight new findings of scientific interest. Maybe a discussion section? Or maybe the tables I suggested at the previous comment could be a synoptic table where differences and similarities from the two sites are highlighted (maybe also replacing long descriptions from the text)?

4a The idea of comparing the effects of the same perturbation system in two very

different places, very distant each other, is fascinating. This should be better stressed and discussed in the manuscript: if the authors go beyond a simple description, this topic has a good research potential.

4b Moreover, the analytic comparison of different EWS is a very interesting and quite unexplored topic in the international literature. To my knowledge, only a few works have been published on this topic (e.g. Baum and Godt, 2010; Lagomarsino et al., 2015; Zezere et al., 2015). This is something that should be stressed by the authors to increase the appeal of their work and to correctly place it in an existing reserach direction that certainly needs to be further expanded. Baum RL, Godt JW (2010) Early warning of rainfall-induced shallow landslides and debris flows in the USA. Landslides 7(3):259–272. doi:10.1007/s10346-009-0177-0 Lagomarsino D, Segoni S, Rosi A, Rossi G, Battistini A, Catani F, Casagli N (2015) Quantitative comparison between two different methodologies to define rainfall thresholds for landslide forecasting. Nat Hazards Earth Syst Sci 15:2413–2423. doi:10.5194/nhess-15-2413-2015 Zêzere JL, Vaz T, Pereira S, Oliveira SC, Marques R, Garcia RAC (2015) Rainfall thresholds for landslide activity in Portugal: a state of the art. Environ Earth Sci, 73(6):2917-2936. doi:10.1007/s12665-014-3672-0

5 A very interesting outcome I saw in your work is that many local or territorial EWS can be integrated in a network, thus providing some sort of a continental or global EWS. Hypothetically, if EWS1 issues an alarm, it could serve as a pre-alarm for EWS2 ... and so on. Your case of study seems to meet this hypothesis. Fom my point of view, this could be a relevant part of the work. Expecially if you connect it with the topicof the lead time (see specific comment.

6 Following my previous comment and comment #3, you could add a figure showing the timeline of the cyclone. That is, a horizontal line representing time, and bars and signs of different colors showing the temporal evolution of: 1- the rainfall event; 2- the effects to the ground in terms of hazard (floods and, above all, landslides); 3- the alarms issued in the two cases of study. Something like that would strengthen the idea

that the two cases of study are deeply linked and an integrated hazard management is desirable.

SPECIFIC COMMENTS

ABSTRACT

The abstract should be revised. Usually abstracts are used to summarize what the article is about. Form this abstract it clearly stands out that this article is about describing an event, not about presenting a relevant research outcome. I suggest to perform all due modifications to the text and then to change the abstract reducing the description of the event and to better stress the research outcomes of the manuscript.

Line 14-15. I think that in Italy every region has a "body" in charge for the hazard assessments. In some regions it is ARPA, in other it may be another institution. Please also note that this kind of detail is better placed in the test site description or EWS description, not in the Abstract.

INTRODUCTION

In my opinion, the authors should identify a gap in the existing state of the art and declare how this work fills the gap and which contribution is provided. Another approach could be to review the literature, to formulate a hypothesis and to check if it is met in the two cases of study.

Line 21 . . .system is operational since 1994. . .

Line 32 "the same meteorological condition" is not exact in my opinion. How you described later, the meteorological conditions varied from Italy to Norway. Maybe "struck by meteorological events belonging to the same perturbation system"?

SECTION 2

LINE 3: I suggest rephrasing with "In particular, Central Europe. . . are sources of . . ."

SECTION 3

Fig.2. In the caption, use Fig. Instead of Figu. And consider explaining that "TO" stands for "Torino".

SECTION 4.2

I think there is a little confusion. I understand that this is a complex system but it is not clear how you combine very different inputs (thresholds, meteo observations, susceptibility maps, models and expert knowledge) before arriving to the hazard assessment.

Line 11. How you observe rainfall conditions? Radars like in the previous test site or rain gauges?

Line 13. Which models?

Line 19. Maybe this is the name of the threshold system or the warning system, rather than the name of the threshold?

Line 21. Which threshold? Can you provide a threshold equation like you did in the previous test site?

Line 31. Twice a day

SECTION 5.2.1

Page 16 lines 7-8: please check the sentence.

SECTION 5.3

Line 8: please check the sentence.

DISCUSSION

The paper misses a discussion of scientific aspects. That should be the core of a research paper. In the discussion section all data gathered and showed in the previous sections should be put together to highlight new findings (if any). Which lessons have

been learnt? How the results of this work can be used by the scientific community? Which advances to the state of the art have been achieved?

CONCLUSIONS

Lines 9-11. To be honest, it is not completely clear to me how the Norwegian forecasting service works. Especially in the Norwegian case study, it is not clear which models have been used, how they are integrated together, what is automated and what is left to expert judgement. Of course, references have been provided, but I think it is better to have a few more details to understand the scientific base of the tools operated here.

Page 26, line 2. "susceptible to that". I suggest rephrasing.

Line 6 "back analysis". The paper does not present a back analysis (i.e. the modeling of a past event), just a description/report of the event.

Lines 8-9. This is one of the most interesting findings of you work. There is a constant struggle in the scientific community to increase the lead time of forecasting systems. Here you show that you can increase it enormously by putting different forecasting services in a network. You should set the stage properly to this sentence, touching this issue in the abstract, in the introduction and in the discussion. Also lines 22-26 are connected with this point.

REFERENCES

This manuscript has a very large number of references. But I see many references that are not "robust" references e.g. they are written in Italian or Norwegian, they are reports or conference abstracts. I suggest limiting this kind of references to the essential ones and to give priority to peer-reviewed articles written in English and published in international journals.

In addition, consider adding references to papers contained in the same special issue and already published in NHESS or in NHESSD: some of them are very well connected with your case of study or with the topic of your manuscript.

---

## Referee Comment (RC3) · Anonymous Referee #3 · 12 Jan 2018

The article describes two landslide warning systems adopted by two regional forecasting services located in Norway and Italy, respectively. The purpose of the work is sharing the experiences from the two different systems, which is an interesting and promising target. However, I have some concerns on the effective scientific contribution of the paper to the NHESS journal readers, at least in its actual form. My impression is that the presentation is not well oriented to a sound scientific analysis. Probably, the submission as a technical note, would be more appropriate.

Please, read in the following my main concerns.

1. The analyzed study cases are interesting. In section 6 the observed events are de-

scribe, as well as the occurred landslides. I wonder why a validation is not presented. I suggest to add an objective evaluation of the system results by using objective metrics, such as AUC, ROC, etc.

2. The two case studies share the same triggering meteorological event. However, the geology and the susceptibility to landslide may be different. I would describe the effects of the same event by emphasizing the similarity and/or the differences of the effects on the two areas and of the two warning systems. Are the types of landslide the same (which actually depend more on the type of soil and slope)? Which are the main differences in terms of areas prone to landslides??This would add more value to the choice of describing these two case studies together.

3. With regard to the Piemonte's landslide forecasting service, 3 different systems are described; however, it is not clear how and if the three systems interact and how the overall service operate. Does the use of one or other model depend on a pre-susceptibility analysis or are they applied to the entire area anyway?

4. Literature review lacks of some contributions in the specific field of early warning system for landslide. I suggest to add some contributions at p2L11. (i.e. Baum and Godt, 2010; Liao et al., 2010; Segoni et al., 2015; Pumo et al., 2016).

Baum R.L. and J W. Godt (2010). Early warning of rainfall-induced shallow landslides and debris flows in the USA. Landslides, 7:259–272 DOI 10.1007/s10346-009-0177-0

S. Segoni, A. Battistini, G. Rossi, A. Rosi, D. Lagomarsino, F. Catani, S. Moretti, and N. Casagli (2015) Technical Note: An operational landslide early warning system at regional scale based on space–time-variable rainfall thresholds. Nat. Hazards Earth Syst. Sci., 15, 853-861

Z. Liao, Y. Hong, J. Wang, H. Fukuoka, K. Sassa, D. Karnawati, and F. Fathani. 2010. Prototyping an experimental early warning system for rainfall-induced landslides in indonesia using satellite remote sensing and geospatial datasets, Landslides.

D. Pumo, A. Francipane, F. Lo Conti, E. Arnone, P. Bitonto, F. Viola, G. La Loggia, L. V. Noto The SESAMO early warning system for rainfall-triggered landslides. Journal of Hydroinformatics. 2016, 18 (2) 256-276; DOI: 10.2166/hydro.2015.060

Please, read in the following my minor comments.

1. P3L2: revise verb of the sentence.

2. P3L17: you could include also April.

3. P5: description of geology is very technical. To facilitate the not geological reader, authors could emphasize the relation of the types of geology domains with the rainfall-triggered landslide (as done in the Norway case). E.g. which are the domains most susceptible to rainfall-triggered landslides?

4. Fig.2a: a map of DEM with hillshade would be more useful and direct. Please improve the quality of the figure and locate Italy in Europe.

5. Fig.2b: to not geological readers, the map of the geological domains does not give information about the propensity to landslides. Actually, it would be preferable to show the slope distribution, which have a clear correlation with landslide, or maybe simplifying the map. Please improve the quality of the figure.

6. Fig. 4: . Please improve the quality of the figure and locate Norway in Europe. Again, DEM and slope distribution would be more helpful than fig.4a.

7. P9L20: remove 'slope phenomena'; it is specified later.

8. P13L3-8: initial conditions are mentioned: please specify at the beginning the selected period for the analysis otherwise it is not clear to what the initial conditions refer to. Description is a bit generic (e.g. "temperature lower than the normal", p13L3). Do you have statistics? Could you quantify?

9. P13L6: is this an annual maximum? Same for p13L10 ("the winter was cold"), too generic.

[Figure]

5. P14, Section 5.2: authors state that the analyzed event had significant impacts in southern Italy. This is in contrast with the choice of Piemonte (northern Italy) as study case! I suggest to modify the sentence.

10. P14L8: In the abstract the period April 27 – May 19 is mentioned. Please, be clear with the selected period.

11. P15L5: Please, specify the duration of recorded rainfall.

12. Is fig.8 important? How about reporting the maxima precipitation daily intensity or the total precipitation?

13. Fig.9: you could add the historical hyetograph measured in one station (e.g. in Turin).

14. Fig.11: please improve quality of figures. Please note that the colors in the map (e.g., light green) do not correspond with those in the legend (e.g. yellow or orange). Is there any transparency effect? Please, solve it, for example, by removing background colors.

---

## Author Comment (AC1) · 19 Jan 2018

We inform that we have read the feedbacks and comments provided by all three reviewers and we would like to thank them for their careful review and their valuable comments. We have appreciated all the comments and suggestions provided. We agree with most of the comments that we considered very constructive and useful to improve the quality of the manuscript. We will address all suggestions preparing a new version of the manuscript. Our replies to general and specific comments of Reviewers #1, #2 and #3 are listed below.

Anonymous Referee #1

[Figure]

General comment / remark:

Overview Two different warning systems against rapid landslide phenomena hazard are shown in present work. The work is, in general, well written but has two main deficiencies: 1) Some details about phenomena and warning thresholds are missing 2) A final comparison between warned phenomena and those occurred is missing

R: We thank Reviewer #1 for point out these deficiencies of the manuscript. We totally agree with the opinion of the Reviewer #1 and we will provided a better explanation in the correspondent chapters. Thank you.

Landslide phenomena "warned" by the two systems seem different. In the case of ARPA Piemonte shallow landslide, deep landslides and channelized debris flows. ARPA Piemonte has three different forecasting approaches: one for each type of phenomenon. In the case of Norwegian Forecasting Service, the "warned" phenomena are: shallow landslide, debris avalanches, slushflows and debris flows. In this case, the writer supposes that debris flow is landslide induced debris flow. Is it right? Then, the writer suggests a section after the introduction where all the "warned" phenomena are briefly introduced and schematized. This section would help the reader to focuse the phenomena, increasing the value of the work.

R: We thank Reviewer #1 for this comment to our manuscript. The landslide phenomena warned by the two systems are the same, mainly shallow landslides and channelized debris flows, but we agree that, this has not been clearly explained in the manuscript. We will provide a new section, in the introduction and as a separate chapter, where we will introduce better the landslide processes that we warn and indicating also the differences in landslide processes in the two regions.

About Norwegian Forecasting Service, there is an unique threshold for all the type of landslides. Authors should justify this.

R: We thank Reviewer #1 for this comment to our manuscript. Yes, we use only one

threshold for all landslide types. We will review the section 4.2, trying to explain better the Norwegian thresholds.

At the end a comparison between launched warnings and occurred phenomena should very useful for understanding the effectiveness of the warning systems. As in the case above, this should increase the value of the work and consequently its diffusion.

R: We thank Reviewer #1 for this comment to our manuscript. We will prepare a figure that will help to compare what has occurred in the two regions and to see issued warnings and occurred landslides, shown by time line as it was also recommended by referee #2.

Specific comments:

Abstract Sentences at lines 14-17 ("In Italy,................in Northwestern Italy.") are redundant. Just write that in Italy the landslide hazard assessment is not national but provided by the Regional Agency for Environmental Protection and that in present paper it is shown the work of ARPA Piemonte.

R: These sentences will be changed and improved as suggested. Thank you.

Sentence at lines 24-27 is too long.

R: This sentence will be divided and split in shorter sentences as it was suggested. Thank you. Introduction

Sentences at lines 7-16 ("The spatial occurrence.........shallow landslides") appears confuse. At the beginning rainfall and snowmelt induced landslide are introduced. After that rapid mass movements are introduced and rainfall and snowmelt induced landslide becomes a subcategory of them. Finally, there are the shallow landslides. Authors should to identify all the phenomena they warn to avoid confusion in the reader.

R: We thank Reviewer #1 for this comment to our manuscript. We will provide a few sentences here that help to introduced better the landslide processes in general, and

how they are classified, also in relation to triggering parameters. The section, presenting the type of landslide warned, should be also improved, since it was too quickly presented. Here in the introduction, or in a separate chapter the types of processes we warn should be better specified.

At line 19, you could insert the reference Thiene et al., (2016).

R: We thank Reviewer #1 for this suggestion. We will analyse this reference and, if relevant, will be included here.

The Vb cyclones About line 14 of page 3, are you sure about" eastern part of the Alps to NorthWest?"; maybe it could be the contrary.

R: This sentence described how the cyclone moves across Europe. However, the use of "Northwest" may have generated confusion since we use the same word when talk about the region Piemonte. This sentence will be checked and revised. Thank you.

The landslide forecasting services in . . . At line 14 of page 9: "is not based only on a threshold" but also on. . . . . . . .

R: This sentence will be changed, as suggested. Thank you.

Piemonte's landslide forecasting service About the use of DEFENSE. What type of weather radar is used? In some cases after using movable weather radar it could be possible reliably estimate the rainfall depth. In a mountain environment in many cases, radar estimates could be not reliable due to the high spatial variability of rainfall and the distance from the fixed weather radar (Germann et al., 2006; Rabiei and Haberlandt, 2015; Bernard et al., 2016). Authors should introduce some details and cautions on the use of weather radar in a mountain environment.

R: We thank Reviewer #1 for this comment to our manuscript. We will provide a detailed description on Piemonte weather radar network.

The Norwegian Forecasting Service This section is a bit confused and should be reor-

ganized and rewritten. Initially, it is stated that the daily assessment is built on thresholds, real time observations and landslide events (occurring during the event?) previous occurred landslides (inventory) and susceptibility maps. It is reasonable that the inventory of landslide be used for determining the threshold. Is not it? Moreover, also runoff is simulated (by which model?): which is the scope if the threshold combine rainfall and snow with the soil saturation degree? In other words, it is better only describe what it is strictly related to the landslide warning. At last most references are not in English language. Therefore, some detail about model could be written in an appendix.

R: We thank Reviewer #1 for this comment to our manuscript. We agree that this section should be improved, giving the opportunity to readers to understand how the system work. The Norwegian national landslide early warning system is a relatively new system that became operational in 2013. We must admit that one of the major deficiency of the Norwegian system is that, for being operational, we have delayed the publication of scientific English articles that describe entirely the method. Only part of it has been described in conference proceedings and in another article (Piciullo et al., 2017). The system recently has been described in another article submitted to the same special issue and it is under discussion and under revision process. We will rearrange the section 4.2, providing a clearer description of the system, models and thresholds describing only what is relevant for landslide warning and providing the correct references, eliminating also some irrelevant ones.

Antecedent condition At line 6 of page 13, it is better 5 days instead of 120 hours.

R: this sentence will be changed and improved as suggested. Thank you.

Piemonte What is ECMWF at line 12 of page 14?

R: We will write down what this abbreviation means. Thank you.

In the sentences of pages 15-16 a big emphasis is addressed to the occurred flood

and the reaching of warning level in the hydrological network. The writer think that this is eccentric respect to the main object of this work, the landslide forecasting. Therefore, the writer suggests the authors to give a brief description of occurred flood and stress the description of occurred phenomena related to the landslide hazard object of forecasting.

R: We thank Reviewer #1 for this comment to our manuscript. We totally agree and we will change this part, as suggested. Thank you

Warning levels The first sentence states that to each warning level corresponds a measure. Is it true? In this case further details will be useful.

R: We thank Reviewer #1 for this comment to our manuscript. Here, we should have used the word "recommended action" instead. We will change the sentence and also table 1, indicating more clearly which general actions are recommended to the stakeholders in the different levels.

About debris flows warned (see line 5 of page 20) which radar was used?

R: We will indicate which radar we have used. Thank you.

Suggestions for references:

R: The referee #1 has recommended the consultation of the following references. We thank Reviewer #1 for these recommended references. We will consult and analyse them and, if relevant, they will be included in the manuscript in the correspondent chapters. • Bernard M., Stancanelli L., Berti M., Simoni A., Gregoretti C., Lanzoni S. (2016) Field results from the runoff generated debris flows occurred at Rovina di Cancia (Venetian Dolomites) XXXV Convegno di Idraulica e Costruzioni Idrauliche – Bologna • Germann, U., Galli, G., Boscacci, M., and Bolliger, M. (2006). Radar precipitation measurement in a mountainous region. Quarterly Journal of the Royal Meteorological Society, 132(618), 1669-1692. http://doi.org/10.1256/qj.05.190 • Rabiei, E., and Haberlandt, U. (2015). Applying bias correction for merging rain gauge and radar data.

Journal of Hydrology, 522, 544-557. http://doi.org/10.1016/j.jhydrol.2015.01.020 •
Thiene, M., Shaw, W. D., and Scarpa R., (2016). Perceived risks of mountain landslides
in Italy: Stated choices for subjective risk reductions. Landslide, doi:10.1007/s10346-
016-0741-3 1.

Anonymous Referee #2

General comment / remark:

I have read and carefully evaluated the manuscript "Regional landslide forecasting
in Piemonte (Italy) and in Norway: experiences from 2013 late spring", submitted to
NHESS by Tiranti et al. The manuscript describes a large cyclone system that struck
two very different and distant test sites (Norway and northern Italy). Plenty of details
are provided about the event, the related hazard and how it was managed in the two
sites. The topic is very interesting and completely centered on the scopes of the journal
and of the special issue.

R: We thank Reviewer #2 for finding this manuscript interesting and for the positive
feedback. However, what we have here is more a couple of (interesting) event reports
than a "real" research paper.

Since NHESS does not consider manuscript types such as reports, technical notes,
event descriptions and so on, my recommendation is to perform "MAJOR REVISIONS"
in order to better highlight and discuss the interesting scientific and research outcomes
that I saw in the manuscript. I want to stress that I consider this manuscript a very
interesting contribution and that it has a very good potential. It just needs to be put in
the right perspective (research-oriented paper instead than an event report). My sub-
sequent comments are aimed to this change of perspective. I encourage the authors to
carry on and adjust the manuscript to make it become a high-quality research article.

R: We thank Reviewer #2 for this comment. The manuscript in our opinion is unique,
because for the first time two operational systems are compared, looking at what hap-

pened and experiences. However, we must admit that we have difficulties to organize the manuscript in a way that we could balance between research and event description. We totally agree that the article, as it is presented, is not a real research article, resembling most to a technical note. We agree that the manuscript needs a major revision if should be presented as research article, and we agree that is worth to try to change it. Therefore, we will carefully review the manuscript according to the many valuable comments and recommendations provided. Thank you.

1 The English is fully understandable and good (although I identified some minor errors). I am not a native speaker, however I suggest to check the text again and especially to split long sentences. They may be hard to follow.

R: We will check the text again and split the long sentences as suggested. Thank you.

2 The manuscript is very rich of details. At the beginning I was positively impressed, but after several pages it got quite boring: the description of the event and of the EWS infrastructures are very good but sometimes the scientific content and the research aspect were missing. And I think those are the main things that NHESS readers are expecting form a research paper. I suggest reducing the description (maybe using a few tables?) and to let research topics stand out from the text (see e.g. my following comments).

R: We thank Reviewer #2 for this comment. We agree that the description could be reduced by using figures and tables, and research topic should be better presented. We will review the manuscript according to the comments and recommendations provided. Thank you.

3 Another issue is that the two cases of study stay always kept separated. There must be a point in the manuscript when things are put together and a synthesis is made to highlight new findings of scientific interest. Maybe a discussion section? Or maybe the tables I suggested at the previous comment could be a synoptic table where differences and similarities from the two sites are highlighted (maybe also replacing

long descriptions from the text)?

R: We thank Reviewer #2 for this comment. Yes, the comparison of the two systems should have been the focus of the paper, and we agree that this comparison was not well done and presented. We will make the changes according to the comments and recommendations provided by reviewers #1, #2 and #3. We will add a figure or table showing differences and similarities and if is necessary a chapter of discussion, depending on the outcomes after a new revision. Thank you.

4a The idea of comparing the effects of the same perturbation system in two very different places, very distant each other, is fascinating. This should be better stressed and discussed in the manuscript: if the authors go beyond a simple description, this topic has a good research potential.

R: We thank Reviewer #2 for these positive feedbacks to our manuscript that gives us the possibility to explain the unique aspect of this article and why we need to look at the secondary hazards (like landslides) due to the similar perturbation system. We have observed in many occasions that the same atmospheric circulation has caused landslides in different countries, however, this has not been described in previous documents and it is not been explored until now. Mainly because operational landslide warnings are relatively new and, traditionally, weather warning and landslide warnings have been left separated. We agree that this unique aspect of the paper should have been better presented in the manuscript. We will change the paper according to the recommendations and suggestions.

4b Moreover, the analytic comparison of different EWS is a very interesting and quite unexplored topic in the international literature. To my knowledge, only a few works have been published on this topic (e.g. Baum and Godt, 2010; Lagomarsino et al., 2015; Zezere et al., 2015). This is something that should be stressed by the authors to increase the appeal of their work and to correctly place it in an existing research direction that certainly needs to be further expanded.

R: We thank Reviewer #2 for these positive feedbacks. Yes, this aspect is a quite unexplored topic and a unique example, as already mentioned in the previous answer. In literature, the comparison of thresholds that have been proposed in different countries or regions, but thresholds are not always used in a operational contest. In few articles the comparison between existing and past systems was also presented. We agree that this literature analysis was not properly described in our manuscript. We will need to summarize better previous similar works and emphasize eventually the lack of similar comparisons and the reasons of that.

5 A very interesting outcome I saw in your work is that many local or territorial EWS can be integrated in a network, thus providing some sort of a continental or global EWS. Hypothetically, if EWS1 issues an alarm, it could serve as a pre-alarm for EWS2 ... and so on. Your case of study seems to meet this hypothesis. Fom my point of view, this could be a relevant part of the work. Expecially if you connect it with the topic of the lead time (see specific comment.

R: We thank Reviewer #2 for these positive suggestions. Yes, this could be the idea. We believe that a network of operational systems of rainfall-induced landslides could be beneficial at European level to share experiences and for a better warning, but we missed to present this idea in our manuscript. We will correct the manuscript, according to the comment.

6 Following my previous comment and comment #3, you could add a figure showing the timeline of the cyclone. That is, a horizontal line representing time, and bars and signs of different colors showing the temporal evolution of: 1- the rainfall event; 2- the effects to the ground in terms of hazard (floods and, above all, landslides); 3- the alarms issued in the two cases of study. Something like that would strengthen the idea

R: We thank Reviewer #2 for these positive suggestions. We agree that we need these figures to improve the quality of the article.

Specific comments:

ABSTRACT The abstract should be revised. Usually abstracts are used to summarize what the article is about. Form this abstract it clearly stands out that this article is about describing an event, not about presenting a relevant research outcome. I suggest to perform all due modifications to the text and then to change the abstract reducing the description of the event and to better stress the research outcomes of the manuscript.

R: The abstract will be changed and improved, after the revision of the manuscript as suggested. Thank you.

Line 14-15. I think that in Italy every region has a "body" in charge for the hazard assessments. In some regions it is ARPA, in other it may be another institution. Please also note that this kind of detail is better placed in the test site description or EWS description, not in the Abstract.

R: This sentence will be review, changed and replaced in another chapter as suggested. Thank you.

INTRODUCTION In my opinion, the authors should identify a gap in the existing state of the art and declare how this work fills the gap and which contribution is provided. Another approach could be to review the literature, to formulate a hypothesis and to check if it is met in the two cases of study.

R: We thank Reviewer #2 for this comment. Yes, we need to review the literature and explain the importance of our contribution compared to the international context.

Line 21 : : :system is operational since 1994: : :

R: This sentence will be changed as suggested. Thank you.

Line 32 "the same meteorological condition" is not exact in my opinion. How you described later, the meteorological conditions varied from Italy to Norway. Maybe "struck by meteorological events belonging to the same perturbation system"?

R: This sentence will review and changed. Thank you.

SECTION 2 LINE 3: I suggest rephrasing with "In particular, Central Europe: : : are sources of : : :"

R: This sentence will be changed and improved as suggested. Thank you.

SECTION 3 Fig.2. In the caption, use Fig. Instead of Figu. And consider explaining that "TO" stands for "Torino".

R: This caption will be corrected as suggested. Thank you.

SECTION 4.2 I think there is a little confusion. I understand that this is a complex system but it is not clear how you combine very different inputs (thresholds, meteo observations, susceptibility maps, models and expert knowledge) before arriving to the hazard assessment.

R: We thank Reviewer #2 for this comment to our manuscript. As answered to reviewer #1, we agree that this section should be improved giving the opportunity to readers to understand how the system work. As already answered to reviewer #1, we cited few articles in English that described entirely the system, but we have described the system in some of them and we need to refer more to them. The system has been recently described in another article, submitted to the same special issue, currently under discussion and under revision. In this article under discussion, we have presented a diagram that explain the process how to arrive to the hazard assessment. Maybe we should refer to this diagram or prepare a simple figure. We will rearrange the section 4.2, providing a clearer description of the system, models and thresholds describing only what is relevant for landslide warning and eliminating irrelevant references.

Line 11. How you observe rainfall conditions? Radars like in the previous test site or rain gauges?

R: We thank Reviewer #2 for this question. We use both! We will explain better this part during the next revision. Thank you.

Line 13. Which models?

R: We thank Reviewer #2 for this question. We use two models: the main model used is a distributed version of the conceptual HBV-model (Beldring et al., 2003) using temperature and precipitation as only input variables. In addition, a physically based model, SoilFlow (Zelalem et al. in preparation), it is run for selected locations as a supplement to the HBV model. This model requires wind speed, air humidity and radiation as input data in addition. Beldring et al., 2003 - Beldring, S., Engeland, K., Roald, L.A., Sælthun, N.R., Voksø, A., 2003. Estimation of parameters in a distributed precipitation-runoff model for Norway. Hydrology and Earth System Sciences. 7: 304-316. We will explain better the use of these models in the next revision of the manuscript. Thank you. Line 19. Maybe this is the name of the threshold system or the warning system, rather than the name of the threshold?

R: This is the name of the threshold, we do not have name for the warning system. We will provide a better explanation. Thank you.

Line 21. Which threshold? Can you provide a threshold equation like you did in the previous test site?

R: We thank Reviewer #2 for this question that allow us to explain better the thresholds in use. The thresholds are used as a strong indicator for increased landslide hazard, at a regional scale. Since the thresholds are based on daily values only, they are not comparable to traditional ID-thresholds. Also, since the thresholds were derived from tree-classification technique, the threshold consists of several linear equations generating thresholds with a latter shape. Currently, Norway is divided into three climate domains, each having individual thresholds, however it is based on the same combination of hydrometeorological conditions (water supply and soil water saturation, fig. 6A). The domains are SE-Norway (dry continental climate), southern-most part of S-Norway (medium wet climate with sparse soil coverage) and the rest of Norway (wet coastal climate). Threshold analysis and development is in progress, and new domains are expected in the future. Also, a paper documenting the work is being prepared. Generally speaking, thresholds indicate increased landslide hazard when values of water supply are higher than 6-8% of the mean annual precipitation combined with a simulated soil water saturation degree higher than 60% (Fig. 6A). The spatial distribution of the thresholds is visualized as raster data (with 1-km2 resolution) at xgeo.no. A regional impact displayed is required to consider issuing a warning. Xgeo.no is a web portal that assist experts in the daily forecast of floods, snow avalanches and landslides (Fig. 6d). Here, prognosis and simulated parameters are daily published for the next six days (Devoli et al., 2014). We will review this section together with this explanation and we will change it, according to the comment.

Line 31. Twice a day

R: This sentence will be changed as suggested. Thank you.

SECTION 5.2.1 Page 16 lines 7-8: please check the sentence.

R: This sentence will be review and changed. Thank you.

SECTION 5.3 Line 8: please check the sentence.

R: This sentence will be review and changed. Thank you.

DISCUSSION The paper misses a discussion of scientific aspects. That should be the core of a research paper. In the discussion section all data gathered and showed in the previous sections should be put together to highlight new findings (if any). Which lessons have been learnt? How the results of this work can be used by the scientific community? Which advances to the state of the art have been achieved?

R: We thank Reviewer #2 for this comment to our manuscript. Yes, we agree that a discussion chapter is needed and the questions proposed should be used to guide us in the redaction of the conclusions and discussions chapters.

CONCLUSIONS Lines 9-11. To be honest, it is not completely clear to me how the Norwegian forecasting service works. Especially in the Norwegian case study, it is not clear which models have been used, how they are integrated together, what is

automated and what is left to expert judgement. Of course, references have been provided, but I think it is better to have a few more details to understand the scientific base of the tools operated here.

R: We thank Reviewer #2 for this comment to our manuscript. Yes, again the Norwegian system is poorly described and we will rearrange the section 4.2, and again here in the conclusions.

Page 26, line 2. "susceptible to that". I suggest rephrasing.

R: This sentence will be changed and improved as suggested. Thank you.

Line 6 "back analysis". The paper does not present a back analysis (i.e. the modeling of a past event), just a description/report of the event.

R: This sentence will be changed according to the comment. Thank you.

Lines 8-9. This is one of the most interesting findings of you work. There is a constant struggle in the scientific community to increase the lead time of forecasting systems.

R: According to the comment, this part need to be better emphasized. We agree, thank you.

Here you show that you can increase it enormously by putting different forecasting services in a network. You should set the stage properly to this sentence, touching this issue in the abstract, in the introduction and in the discussion. Also lines 22-26 are connected with this point.

R: Yes, we will review the main issue/contribution of the paper. We agree, thank you. Suggestions for references:

REFERENCES This manuscript has a very large number of references. But I see many references that are not "robust" references e.g. they are written in Italian or Norwegian, they are reports or conference abstracts. I suggest limiting this kind of references to the essential ones and to give priority to peer-reviewed articles written in English and

published in international journals.

R: We thank Reviewer #2 for this comment. Common problems for most of the operational systems is that for being operational, they lack time to publish articles on international journals. However, we believe that is better to present some important references even if they are in Norwegian or Italian, especially those technical reports, instead of do not have references at all, but we agree that we should eliminate those references that are not robust.

In addition, consider adding references to papers contained in the same special issue and already published in NHESS or in NHESSD: some of them are very well connected with your case of study or with the topic of your manuscript. Baum RL, Godt JW (2010) Early warning of rainfall-induced shallow landslides and debris flows in the USA. Landslides 7(3):259–272. doi:10.1007/s10346-009-0177-0 Lagomarsino D, Segoni S, Rosi A, Rossi G, Battistini A, Catani F, Casagli N (2015) Quantitative comparison between two different methodologies to define rainfall thresholds for landslide forecasting. Nat Hazards Earth Syst Sci 15:2413–2423. doi:10.5194/nhess-15-2413-2015 Zêzere JL, Vaz T, Pereira S, Oliveira SC, Marques R, Garcia RAC (2015) Rainfall thresholds for landslide activity in Portugal: a state of the art. Environ Earth Sci, 73(6):2917-2936. doi:10.1007/s12665-014-3672-0

R: We thank Reviewer #2 for these recommended references. They can be very relevant. We will consult and analyse them and will be included in the manuscript, in the correspondent chapters, where is needed.

Anonymous Referee #3

General comment / remark: The article describes two landslide warning systems adopted by two regional forecasting services located in Norway and Italy, respectively. The purpose of the work is sharing the experiences from the two different systems, which is an interesting and promising target. However, I have some concerns on the effective scientific contribution of the paper to the NHESS journal readers, at least in

its actual form. My impression is that the presentation is not well oriented to a sound scientific analysis. Probably, the submission as a technical note, would be more appropriate.

R: We thank Reviewer #3 for this positive comment. We agree that in the present form the article is closer to a technical note than a research article. Based on these comments and those from reviewer #1 and #2 we will change the manuscript emphasizing research topics, not clearly presented.

Please, read in the following my main concerns. 1. The analyzed study cases are interesting. In section 6 the observed events are describe, as well as the occurred landslides. I wonder why a validation is not presented. I suggest to add an objective evaluation of the system results by using objective metrics, such as AUC, ROC, etc.

R: We thank Reviewer #3 for finding this case interesting. We agree that a section, in which the systems and cases are better compared, and results validated, is missing. However, the main focus of the paper is to present that the same weather system caused landslides in two different regions in Europe and experience with the warning messages issued, by two quite similar forecasting services. More extensive validation of impacts (type of landslides and economic consequences) could be part of another article.

2. The two case studies share the same triggering meteorological event. However, the geology and the susceptibility to landslide may be different. I would describe the effects of the same event by emphasizing the similarity and/or the differences of the effects on the two areas and of the two warning systems. Are the types of landslide the same (which actually depend more on the type of soil and slope)? Which are the main differences in terms of areas prone to landslides??This would add more value to the choice of describing these two case studies together.

R: We thank Reviewer #3 for this comment. Yes, we agree that the geological, topographic similarities and differences of the two regions, also in terms of landslide

processes could have been better presented and eventually discussed the susceptibility of the regions instead of a detailed geological description. We will go through the manuscript and evaluate how this section could be improved. In addition, we will insert a section to explain better similarities in the landslide processes and warning systems, according also to the comments from reviewer#1.

3. With regard to the Piemonte's landslide forecasting service, 3 different systems are described; however, it is not clear how and if the three systems interact and how the overall service operate. Does the use of one or other model depend on a presusceptibility analysis or are they applied to the entire area anyway?

R: We thank Reviewer #3 for this comment. Yes, we will rephrase providing clearer description.

4. Literature review lacks of some contributions in the specific field of early warning system for landslide. I suggest to add some contributions at p2L11. (i.e. Baum and Godt, 2010; Liao et al., 2010; Segoni et al., 2015; Pumo et al., 2016).

R: We thank Reviewer #3 for the recommended references. Some of these were also recommended by the other two reviewers. We will consult them and we insert them in the correspondent chapters, where is needed.

Specific comments:

Please, read in the following my minor comments. 1. P3L2: revise verb of the sentence.

R: The verb in this sentence will be changed, according to the comment. Thank you.

2. P3L17: you could include also April.

R: We will add April in this sentence as suggested. Thank you.

3. P5: description of geology is very technical. To facilitate the not geological reader, authors could emphasize the relation of the types of geology domains with the rainfall triggered landslide (as done in the Norway case). E.g. which are the domains most

susceptible to rainfall-triggered landslides?

R: We thank the reviewer#3 for this comment. We agree that for the focus of the manuscript a detailed geological description is not necessary. We will evaluate during the second revision if it could be more useful to present a map of landslides domains (or susceptibility maps), showing within the regions, which areas are more exposed to landslides. This section will be updated according to the comment. Thank you.

4. Fig.2a: a map of DEM with hillshade would be more useful and direct. Please improve the quality of the figure and locate Italy in Europe.

R: This figure will be changed and improved as suggested. Thank you.

5. Fig.2b: to not geological readers, the map of the geological domains does not give information about the propensity to landslides. Actually, it would be preferable to show the slope distribution, which have a clear correlation with landslide, or maybe simplifying the map. Please improve the quality of the figure.

R: We thank the reviewer#3 for this comment. We agree that a map with slope angle or hillshade can show better the geomorphological and topographic conditions of the regions.

6. Fig. 4: . Please improve the quality of the figure and locate Norway in Europe. Again, DEM and slope distribution would be more helpful than fig.4a.

R: This figure will be changed and improved as suggested. We will also include a map of Europe showing the location of the study area. We agree that a map with slope angle or hill-shaded can show better the geomorphological and topographic conditions of the region.

7. P9L20: remove 'slope phenomena'; it is specified later.

R: This sentence will be changed and improved as suggested. Thank you.

8. P13L3-8: initial conditions are mentioned: please specify at the beginning the se-

lected period for the analysis otherwise it is not clear to what the initial conditions refer to. Description is a bit generic (e.g. "temperature lower than the normal", p13L3). Do you have statistics? Could you quantify?

R: We thank the reviewer#3 for this comment. We agree that this part was quickly presented. We will specify better the antecedent period. The idea with giving some general information on the temperature as a start is to provide some background for the development of the large-scale weather system. Values vary within the region, if found it necessary we can provide a range. However, we do agree that more specific values must be provided.

9. P13L6: is this an annual maximum? Same for p13L10 ("the winter was cold"), too generic.

R: This sentence will be review and changed as suggested. For the second comment, we agree that the sentence is too generic. We will correct according to the comment. Thank you.

5. P14, Section 5.2: authors state that the analyzed event had significant impacts in southern Italy. This is in contrast with the choice of Piemonte (northern Italy) as study case! I suggest to modify the sentence.

R: In this sentence we were are referring to the impacts in the initial phase of the Vb cyclone, mainly sandstorm and strong wind (which were dominant in southern Italy and Malta), but the impacts due to intense rainfalls were more significant in Piemonte and Norway. We agree that these sentences should be rephrased, according to the comment. Thank you.

10. P14L8: In the abstract the period April 27 – May 19 is mentioned. Please, be clear with the selected period.

R: The period should be corrected to be the same in the manuscript and abstract. Thank you.

11. P15L5: Please, specify the duration of recorded rainfall.

R: The duration will be specified in this sentence and changed, as suggested. Thank you.

12. Is fig.8 important? How about reporting the maxima precipitation daily intensity or the total precipitation? 13. Fig.9: you could add the historical hyetograph measured in one station (e.g. in Turin).

R: After also comments received from reviewer #2 we will provide another figure showing surface maps of the evolution of the system across Europe. We will evaluate eventually the possibility to add 2 figures showing the total precipitation or max precipitation in two selected meteorological stations in Piemonte and Norway as reviewer #3 suggest. Thank you.

14. Fig.11: please improve quality of figures. Please note that the colors in the map (e.g., light green) do not correspond with those in the legend (e.g. yellow or orange). Is there any transparency effect? Please, solve it, for example, by removing background colors.

R: The figure will be improved, according to the comments. Thank you.

Suggestions for references:

Baum R.L. and J W. Godt (2010). Early warning of rainfall-induced shallow landslides and debris flows in the USA. Landslides, 7:259–272 DOI 10.1007/s10346-009-0177-0 S. Segoni, A. Battistini, G. Rossi, A. Rosi, D. Lagomarsino, F. Catani, S. Moretti, and N. Casagli (2015) Technical Note: An operational landslide early warning system at regional scale based on space–time-variable rainfall thresholds. Nat. Hazards Earth Syst. Sci., 15, 853-861 Z. Liao, Y. Hong, J. Wang, H. Fukuoka, K. Sassa, D. Karnawati, and F. Fathani. 2010. Prototyping an experimental early warning system for rainfall-induced landslides in Indonesia using satellite remote sensing and geospatial datasets, Landslides. D. Pumo, A. Francipane, F. Lo Conti, E. Arnone, P. Bitonto, F. Viola, G. La

Loggia, L. V. Noto The SESAMO early warning system for rainfall-triggered landslides. Journal of Hydroinformatics. 2016, 18 (2) 256-276; DOI: 10.2166/hydro.2015.060

R: We thank Reviewer #3 for these recommended references. They can be very relevant. We will consult and analyse them and if relevant, will be included in the manuscript, in the correspondent chapters.
* * *

---

## Referee Report (RR1)

**RE:** NHESS 2017 411         Tiranti et al. Comparison of landslide forecasting services in Piemonte (Italy) and in Norway, illustrated by events in late spring 2013.

**Overview**

The authors improved the paper that now is easily readable. However, there are two points to be exploited/discussed:

1) A final comparison between warned phenomena and those occurred is missing for the Norway territory hit by the cyclon: they show the map of occurred phenomena in Figure 11 and the warnings in Figure 14. A new figure showing warned area and locations of the occurred events should be provided as done in figure 17. After that, the value of POD and FAR (see page 30 of the submitted manuscript) should be computed.

2) About the EWS of Norway, some simple explanation should be addressed about the use of the same threshold for all the phenomena: shallow landslide, debris avalanches, slushflows and debris flows.

Suggestion: values of POD and FAR should be used to strengthen the outcomes in the conclusions.

The following are the detailed comments and specifications.

**Introduction**

At page 2, line 5 "of and"???; line 31 the reference Hungr et al (2014) should be between bracktes. At page 3, line 1 perhaps it is better "volume inferior to 5000 m$^3$", line12 perhaps it is better "tens of minutes" instead of "minutes"; line 32 "shortage of personnel" it is better than "loss of personnel". At page 4, lines 16-17 the sentence does not sound; line 24 large-scale pattern of what????. Moreover, as the authors study the hazard phenomena triggered by synoptic scale meteorological processes, the writer suggests the reading of the work of Underwood et al (2016) that studied the meteorological processes associated to the convective rainfalls that triggered debris flows on Italian Alps.

**The landslide forecasting service in Piemonte region and Norway**

Table 1: as threshold parameters "the rainfall from rain gauge" is setted for Piemonte, Italy columns. This seem to contradict the type of thresholds, where radar hourly intensity rainfall threshold is used. Perhaps, it is rainfall, the threshold parameter. Moreover, I suggest the authors to change the threshold parameters into threshold quantities.

General comment: it seems that this EWS warns the channelized debris flows occurring after large-scale meteoric processes. Unfortunately on summer debris flow of large magnitude could occur after local high intensity convective rainfall: the case of Rovina di Cancia where the 23th of July a debris flow was triggered by an isolated precipitation of about 30 mm in half of hour, concentrated in about 1 square kilometer while in the surrounding area for an extension of more than 50 square kilometer no rainfall was recorded according to Gregoretti et al. (2016). Therefore, authors should add a sentence where it is specified that channelized debris flows caused by isolated intense convective rainfall are not considered by the present EWSs.

At page 12, I would suggest some more details on the areas where radar visibility is good. How much is it deep starting from the mountains feet?

Page 14, line 10: compares perhaps it is better than consults; page 15, lines 12-13, perhaps there is a contradiction: how a hydrological model that use precipitation as input variable can predict rainfall? Line 28, latter shape is it correct?

**Meteorological conditions in late spring 2013**

At page 21, line 9: please substitute Saturday and Sunday with the day number. At page 23, line 14 most of????. At page 24 line 13 100-year return period of what? Flood?. The writer think it could be the peak flood discharge. Therefore, specify it and also at lines 14-18

Gregoretti, C., Degetto, M., Bernard, M., Crucil, G., Pimazzoni, A., De Vido, G., Berti, M., Simoni, A., Lanzoni, S., (2016a). Runoff of small rocky headwater catchments: Field observations and hydrological modeling. Water Resources Research. http://doi.org/10.1002/2016WR018675

Underwood S.J., Schultz M.D., Berti M., Gregoretti, C., Simoni A., Mote T.L., Hayser A., and Saylor A.M. (2016) Atmospheric circulation patterns, cloud-to-ground lightning, and locally intense convective rainfall associated with debris flow initiation in the Dolomite Alps of northeastern Italy. Natural Hazard and Earth System Science, 16, 509-528, doi:10.5194/nhess-16-509-2016

---

## Referee Report (RR2)

[referee-annotated manuscript omitted]

---

## Author Response (AR2)

Editor Decision: Publish subject to minor revisions (review by editor) (04 Apr 2018) by Stefano Luigi Gariano
Comments to the Author:
Dear Davide Tiranti and co-authors,
the second check made by two reviewers highlighted that you have improved considerably your article, as to being acceptable for publication.
However, they provided few comments and some corrections, useful for a final amelioration of your work.
Moreover, in the following, I list some other technical corrections, mostly regarding tables and figures, proposed by myself.
After addressing these corrections, your manuscript will be briefly checked again by me and then will proceed to proofreading for the final publication in NHESS journal, in the selected Special Issue.
Best regards.
Sincerely,
Stefano Luigi Gariano
NHESS Guest Editor

**Re: Dear Editor,**
**Thank you for suggested reviews, that we addressed.**
**We changed the First and Corresponding Author (now Graziella Devoli), please refer to the new one since now.**
**Best regards,**
**Davide Tiranti**

Technical corrections
Figure 1B: Please check the legend of slope. I suggest using non-overlapping classes instead of limits. As an example, if a pixel has a slope equal to 10 degrees, it is in the second or in the third class? The same for values equal to 20, 30, 40, and 50.
**Re: Done.**
Figure 1C: Please check the legend of shallow landslide density. I suggest using non-overlapping classes for values higher than 5. Please check the unit of measurement.
**Re: Done.**
Figure 2: Please correect the legend by using non-overlapping classes instead of (upper or lower?) limits.
**Re: Done.**
Figure 4: Please correct the legend by using non-overlapping classes and including or excluding uppper and lower limits. Please change "Annual mean precipitation" into "Mean annual precipitation" or "MAP". Also in the caption please change "Annual Precipitation Map" into "Mean Annual Precipitation".
**Re: Done.**
Figure 8: Please check the legend. I suggest using non-overlapping classes.
**Re: Done.**
Figure 9: Please check the legend. The measuring resolution is of the order of 1 mm?
**Re: The Figure 9 shows mean sea level pressure (Pa) analysis over Europe on 16th May 2013 at 00:00 UTC derived by numerical weather prediction model operated by ECMWF.**
Figure 12: Please correct "km2" in the legend, by using superscript.
**Re: Done.**
Figure 17: I suggest adding a legend.
**Re: We added in the figure caption.**
Figures 1, 3, 5, 6, 8, 11, 13, 14, 16: These figures have a low quality in the pdf file. Probably, this is due to the compression process. However, I suggest checking the readability of all figures, in particular regarding labels and legends.
**Re: Yes, it is due to the pdf compression. However, the original figures have an appropriated resolution.**
Table 1. I suggest using always two columns also for equal values, e.g. for "Operative (daily)", "Rainfall and snowmelt", and "4".
**Re: Done.**
I suggest correct "rainfalls" by using "rainfall" (uncountable) everywhere in the text.

**Re: Done.**
Please use the abbreviation "Fig." when it appears in running text or in brackets, unless it comes at the beginning of a sentence (see https://www.natural-hazards-and-earth-systemsciences. net/for_authors/manuscript_preparation.html).
Please do not use abbreviations for Table.
**Re: Done.**
Please add DOI to each reference in the list, where available. Please check the style of the reference list, following the Copernicus guidelines (https://www.natural-hazards-and-earth-systemsciences. net/Copernicus_Publications_Reference_Types.pdf)
**Re: Done.**

**Dear Reviewers,**
**thank you for your precious contribution.**
**We modified the manuscript according with your suggestions, when possible.**
**Best regards,**
**The Authors**

RE: NHESS 2017 411 Tiranti et al. Comparison of landslide forecasting services in Piemonte (Italy) and in Norway, illustrated by events in late spring 2013.
Overview
The authors improved the paper that now is easily readable. However, there are two points to be exploited/discussed:
1) A final comparison between warned phenomena and those occurred is missing for the Norway territory hit by the cyclon: they show the map of occurred phenomena in Figure 11 and the warnings in Figure 14. A new figure showing warned area and locations of the occurred events should be provided as done in figure 17. After that, the value of POD and FAR (see page 30 of the submitted manuscript) should be computed.
**Re: Thank you for your suggestions, we tried to satisfy all the requests compatibly with the maximum paper-length allowed.**
2) About the EWS of Norway, some simple explanation should be addressed about the use of the same threshold for all the phenomena: shallow landslide, debris avalanches, slushflows and debris flows.
**Re: Thank you. Explanation of use of thresholds is already in the text.**
The following are the detailed comments and specifications.
Introduction
At page 2, line 5 "of and"???; line 31 the reference Hungr et al (2014) should be between bracktes. At page 3, line 1 perhaps it is better "volume inferior to 5000 m3", line12 perhaps it is better "tens of minutes" instead of "minutes"; line 32 "shortage of personnel" it is better than "loss of personnel". At page 4, lines 16-17 the sentence does not sound; line 24 large-scale pattern of what????. Moreover, as the authors study the hazard phenomena triggered by synoptic scale meteorological processes, the writer suggests the reading of the work of Underwood et al (2016) that studied the meteorological processes associated to the convective rainfalls that triggered debris flows on Italian Alps.
**Re: Corrections done.**
The landslide forecasting service in Piemonte region and Norway
Table 1: as threshold parameters "the rainfall from rain gauge" is setted for Piemonte, Italy columns. This seem to contradict the type of thresholds, where radar hourly intensity rainfall threshold is used. Perhaps, it is rainfall, the threshold parameter. Moreover, I suggest the authors to change the threshold parameters into threshold quantities.
**Re: Corrections done.**
General comment: it seems that this EWS warns the channelized debris flows occurring after large-scale meteoric processes. Unfortunately on summer debris flow of large magnitude could occur after local high intensity convective rainfall: the case of Rovina di Cancia where the 23th of July a debris flow was triggered

by an isolated precipitation of about 30 mm in half of hour, concentrated in about 1 square kilometer while in the surrounding area for an extension of more than 50 square kilometer no rainfall was recorded according to Gregoretti et al. (2016). Therefore, authors should add a sentence where it is specified that channelized debris flows caused by isolated intense convective rainfall are not considered by the present EWSs.

**Re: Thank you for suggestion, but that's already clarified in Tiranti et al., 2014.**

At page 12, I would suggest some more details on the areas where radar visibility is good. How much is it deep starting from the mountains feet?

**Re: Please, see cited literature.**

Page 14, line 10: compares perhaps it is better than consults; page 15, lines 12-13, perhaps there is a contradiction: how a hydrological model that use precipitation as input variable can predict rainfall? Line 28, latter shape is it correct?

**Re: Corrections done.**

Meteorological conditions in late spring 2013

At page 21, line 9: please substitute Saturday and Sunday with the day number. At page 23, line 14 most of????. At page 24 line 13 100-year return period of what? Flood?. The writer think it could be the peak flood discharge. Therefore, specify it and also at lines 14-18

**Re: Corrections done.**